# An Igh distal enhancer modulates antigen receptor diversity by determining locus conformation

Khalid H. Bhat [1,5,12], Saurabh Priyadarshi[1,12], Sarah Naiyer [1], Xinyan Qu[1,6], Hammad Farooq [2], Eden Kleiman[3,7], Jeffery Xu[3,8], Xue Lei [2,9], Jose F. Cantillo[1,10], Robert Wuerffel[1,11], Nicole Baumgarth[4], Jie Liang [2], Ann J. Feeney[3] & Amy L. Kenter [1] ✉

The mouse Igh locus is organized into a developmentally regulated topologically associated domain (TAD) that is divided into subTADs. Here we identify a series of distal $V_H$ enhancers ($E_{VH}$s) that collaborate to configure the locus. $E_{VH}$s engage in a network of long-range interactions that interconnect the subTADs and the recombination center at the $D_HJ_H$ gene cluster. Deletion of $E_{VH}1$ reduces V gene rearrangement in its vicinity and alters discrete chromatin loops and higher order locus conformation. Reduction in the rearrangement of the $V_H11$ gene used in anti-PtC responses is a likely cause of the observed reduced splenic B1 B cell compartment. $E_{VH}1$ appears to block long-range loop extrusion that in turn contributes to locus contraction and determines the proximity of distant $V_H$ genes to the recombination center. $E_{VH}1$ is a critical architectural and regulatory element that coordinates chromatin conformational states that favor V(D)J rearrangement.

Progenitor B cells must develop a diverse antibody receptor repertoire to provide protection against a wide range of antigens and pathogens. Each mature B cell has a unique Ig receptor, created via V(D)J recombination through an ordered set of rearrangements. One of the ~100 functional Igh locus $V_H$ genes must recombine with a rearranged $DJ_H$ element, which itself is assembled from one of 8–12 $D_H$ and one of 4 $J_H$ gene segments in each pro-B cell[1]. The intergenic control region 1 (IGCR1) separates the $D_H J_H$ clusters from the proximal $V_H$ segments and creates an insulated domain enforcing ordered D to $J_H$ followed by $V_H$ to $DJ_H$ recombination[2–4]. Igh locus contraction[5–7], diffusion related mechanisms[8], and RAG scanning[9] bring $V_H$ segments into closer spatial proximity to the RC permitting usage of the $V_H$ from across the locus.

Nevertheless, the contribution of specific regulatory elements to locus conformation and the rearrangement of distal $V_H$ segments remains unclear.

Chromatin is organized at the Mb scale into topological associating domains (TADs) that encompass spatial neighborhoods of high-frequency chromatin interactions[10–13] that include contact loops anchored by pairs of convergent CTCF binding elements (CBEs) in association with cohesin[14,15]. TAD organization, CBEs, and loop domains may reflect the functional partition of chromatin regions by physiological activities[10–12,14,16]. In this regard, it is significant that the Igh locus is contained within a 2.9-Mb TAD divided into three sub-TADs in pro-B cells[17]. TADs are frequently anchored by motifs bound by the

[1]Department of Microbiology and Immunology, University of Illinois College of Medicine, Chicago, IL 60612-7344, USA. [2]Department of Bioengineering, University of Illinois Colleges of Engineering and Medicine, Chicago, IL 60612-7344, USA. [3]Department of Immunology and Microbiology, IMM-22, Scripps Research, La Jolla, CA 92037, USA. [4]W. Harry Feinstone Dept. Molecular Microbiology and Immunology, Bloomberg School of Public Health, Johns Hopkins University, Baltimore, USA. [5]Present address: SKUAST Kashmir, Division of Basic Science and Humanities, Faculty of Agriculture, Wadura Sopore-193201, Wadoora, India. [6]Present address: Medpace, Cincinnati, Ohio 45227, USA. [7]Present address: Crown Bioscience, San Diego, CA 92127, USA. [8]Present address: Brookwood Baptist Health General Surgery Residency, Birmingham, AL 35211, USA. [9]Present address: Sanford Burnham Prebys Medical Discovery Institute, La Jolla, CA 92037, USA. [10]Present address: Immunotek, S.L. Alcala de Henares, Spain. [11]Present address: 10441 Circle Dr. Apt 47C, Oak Lawn, IL 60453, USA. [12]These authors contributed equally: Khalid H. Bhat, Saurabh Priyadarshi. ✉e-mail: star1@uic.edu

architectural protein CTCF and its interaction partner, cohesin[14,18]. The observation that CBEs situated at TAD boundaries are often in a convergent orientation[14] has led to the proposition that chromatin loops are formed by an extrusion mechanism mediated by cohesin[19–22].

More than 100 CBEs are distributed throughout the locus and are convergent with 10 CBEs (termed 3′ CBE) at the 3′ TAD boundary[2,23]. IGCR1 is composed of a pair of divergent CBEs where one motif is convergent with $V_H$ CBE and the second is convergent with the 3′CBE[24] setting the stage for extensive IGCR1 centric loop extrusion. Studies have implicated loop extrusion in generating Igh locus contraction[25,26] and as a mechanism for a RAG scanning model implicated in VDJ gene assembly[9]. Nevertheless, the Igh locus topology is configured by large chromatin loops that provide access of the $D_H$-distal and -proximal $V_H$ gene segments with rearranged 3′ $D_H J_H$ in pro-B cells[17,27]. These observations raise a question regarding the interplay of loop extrusion and the contribution of locus topology determined by cis-regulatory elements to $V_H$ choice during $V_H$ to $DJ_H$ rearrangement.

We report the discovery of four Igh $V_H$ enhancers ($E_{VH}$1-4). Deletion of $E_{VH}$1 has a structural and functional impact on the Igh locus in cell lines and mice. $E_{VH}$1 participates in an enhancer interactome, determines locus topology, and modulates Igh gene transcription and regional $V_H$ gene usage. Our findings reveal that $E_{VH}$1 may constrain loop extrusion and thereby determines chromatin conformational states that influence $V_H$ to $DJ_H$ recombination.

## Results

### Four enhancers identified in the Igh locus

The Igh TAD is divided into three sub-TADs linked by chromatin interactions between Site I of sub-TAD A, Friend of Site Ia (FrOStIa) and FrOStIb of sub-TAD B, and Sites II, II.5 and III all in sub-TAD C[17] (Fig. 1a, g). To define motifs that anchor Igh chromatin loops, Site I and FrOStIa were visually scanned for distinct transcriptional and epigenetic features. Site I contains the highly transcribed $V_H$14-2 gene promoter (Pr)[28] that is prominently decorated with histone 3 lysine 4 methyl 1 (H3K4me1) and H3K4me3. The presence of H3K4me3 which mark transcriptionally active Prs[29] and the relative absence of enhancer-associated H3K27Ac modifications, indicates that this element predominantly functions as a Pr (Supplementary Fig. 1a). It is also bound by the co-activator MED1 and TFs Pax5 and IRF4 and lacks CTCF and RAD21 (Fig. 1b). We also identified $E_{VH}$1, located in FrOStIa, which is marked by H3K27Ac and H3K4me1 and is bound by TFs PAX5, IRF4, IKAROS, E2A, PU.1, and transcriptional co-activators MED1, BRG1, and p300 (Fig. 1b). H3K4me1 is associated with active and poised enhancers while H3K27Ac distinguishes active enhancers (reviewed in[30]). $E_{VH}$1 is flanked by a CBE that is co-bound by CTCF and RAD21 (Fig. 1b). Likewise, we visually identified a series of other $E_{VH}$s including $E_{VH}$2, $E_{VH}$3, and $E_{VH}$4 (red rectangles) with an epigenetic and TF binding profile similar to $E_{VH}$1 that are interspersed among the intermediate and distal $V_H$ gene exons (Fig. 1a, c)(Supplementary Fig. 1). $E_{VH}$2 is located within FrOStIb in sub-TAD B, and $E_{VH}$3 and $E_{VH}$4 are positioned between Site II and Site II.5 in sub-TAD C (Fig. 1a, c). We consider the possibility that $V_H$14-2 Pr and the $E_{VH}$s play a role in $V_H$ gene transcription, locus topology, and rearrangement of nearby $V_H$ genes.

### Identification of Site I and FrOStIa loop anchors

We examined the spatial organization of Site I and FrOStIa in CD19+ Rag2$^{-/-}$ pro-B cells, the Abelson transformed (Abl-t) Rag1 deficient pro-B cell line 445.3, and ConA activated splenic T cells using the 3C anchors, FrOStIa F.6 and $E_{VH}$1 (Fig. 1d) (Supplementary Fig. 2a, b). The F.6 3C fragment contains a CBE and is located ~15 kb upstream of $E_{VH}$1 (Fig. 1b). In both cases, looping interactions with 3C fragments Site I.1, I.2 and I.3 ($V_H$14-2 gene) were elevated in pro-B cell specific chromatin templates as compared to that found in T cells (Fig. 1d) (Supplementary Fig. 2b). Likewise, the Site I.3 anchor probe, which harbors the $V_H$14-2 gene, associates with F.2-F.3 ($E_{VH}$1) and F.6 (CBE)

and F.8-F.9 in pro-B cell specific fashion (Fig. 1e). Thus, the Site I.3 fragment is a major interaction partner with $E_{VH}$1 and F.6 (CBE). It was difficult to determine candidate loop anchors in Sites I.1 and I.2 due to an unremarkable epigenetic landscape and TF binding profile.

To determine whether the Site I.3 fragment also associates with $D_H$ proximal $V_H$ genes we analyzed locations close to $V_H$ segments (P.1-P.6) and IGCR1 in chromatin from Rag2$^{-/-}$ pro-B cells, the 445.3 line, and ConA activated splenic T cells in 3C assays (Fig. 1f). IGCR1, composed of two CBEs in divergent orientation, is a boundary element that separates the $D_H$-$J_H$ clusters from the most proximal $V_H$ genes[3,4,31] (Fig. 1a). Site I.3 contacts P.1-P.5 and IGCR1 but not P.6 in pro-B cell specific chromatin and not in T cells (Fig. 1f). Thus, Site I.3 engages in a multiplicity of interactions spanning sub-TAD A (blue arc) and may directly or indirectly bridge IGCR1 and the $D_H$ proximal $V_H$ genes with $E_{VH}$1/F.6 located in subTAD B (Fig. 1g).

### Deletion of a $V_H$ promoter, $E_{VH}$ or CBE diminish looping and transcription

To establish a cell culture model to study the involvement of specific transcriptional elements (TE) and CBE motifs in Igh locus function we employed genome editing to generate identical deletions on both alleles of the $V_H$14-2 promoter (Pr), $E_{VH}$1, F.6 CBE, $E_{VH}$2 and Site I.1 in the 445.3.11 sub-line (Fig. 2a)(Supplementary Fig. 3a–d). $V_H$ Prs are composed of a polypyrimidine tract, heptamer and octamer and deletion of the octamer/heptamer is sufficient to abolish Pr activity[32]. CTCF binding at the F.6 CBE in the control and Site I.1 CBE KO lines was detected whereas binding was abolished in the F.6 CBE KO lines in ChIP assays (Fig. 2b). Because there was little epigenetic guidance to identify potential loop anchor motifs within Site I.1 we arbitrarily constructed deletions centered on the $V_H$2-8 exon and an adjacent CBE (Supplementary Fig. 3e). Site I.3:F.6 looping interactions were dependent on the integrity of the $V_H$14-2 Pr, F.6 CBE and $E_{VH}$1 motifs but not on $E_{VH}$2 or deletions around the $V_H$2-8 gene in Site I.1 (Fig. 2c) (Supplementary Fig. 2c, d). F.6:Site I.1 and Site I.3:Site I.1 contacts required intact F.6 CBE and $V_H$14-2 Pr, respectively, indicating the autonomy of each these interactions (Fig. 2c). Thus, the $V_H$14-2 Pr, $E_{VH}$1 and F.6 CBE coordinate Site I:FrOStIa looping.

To validate $E_{VH}$1 enhancer function we tested the transcriptional effects of deletion on $V_H$ germline transcript (GLT) expression. These GLTs are ncRNAs that are transcribed from $V_H$ genes poised to undergo V(D)J rearrangement at the pro-B cell stage. Unlike most $V_H$ GLTs, the high level of $V_H$14-2 GLT expression was evident in the absence of STI-571 induction that is used to induce Ig gene transcription[33]. $V_H$14-2 GLTs were abolished or significantly diminished in the $V_H$14-2 Pr-, $E_{VH}$1- and F.6-KO lines, respectively, as compared to the control demonstrating that the transcription of this GLT is both Pr- and $E_{VH}$1- dependent (Fig. 2d). We studied two additional $V_H$ genes in 445.3.11 cells that were arrested in G1 phase by STI-571 treatment. $V_H$81X is a member of the $V_H$7183 family and the first functional $D_H$-proximal $V_H$ gene and $V_H$2-5 is a member of the $V_H$Q52 family ($V_H$Q52.7.18) and located in sub-TAD A (Fig. 1a). $V_H$ GLTs, composed of a 5′ leader sequence and $V_H$ exon are often found as unspliced and spliced (Fig. 2e). The $V_H$81X (unspliced, 300 bp; spliced, 163 bp) and $V_H$2-5 (unspliced, 338 bp) GLTs are clearly present in the 445.3.11 control and in the $E_{VH}$2 KO lines (Fig. 2e). However, expression of these transcripts is abolished in $E_{VH}$1 KO cells, and diminished in $V_H$14-2 Pr- or F.6_CBE KO lines indicating that 1) $E_{VH}$1 is essential for transcription of at least several $V_H$ genes and 2) that the $V_H$14-2 Pr and F.6 CBE elements may contribute to $E_{VH}$1 function by facilitating its spatial proximity to $V_H$ exons in subTAD A.

### V(D)J recombination requires specific loop anchors in Abl-t lines

To investigate the influence of TE and CBE KOs on V(D)J recombination we studied D- > $J_H$ and $V_H$- > $DJ_H$ rearrangements in Rag1$^{-/-}$ Abl-t 445.3.11 control and KO lines. D- > $J_H$ recombination reflects the activity of the

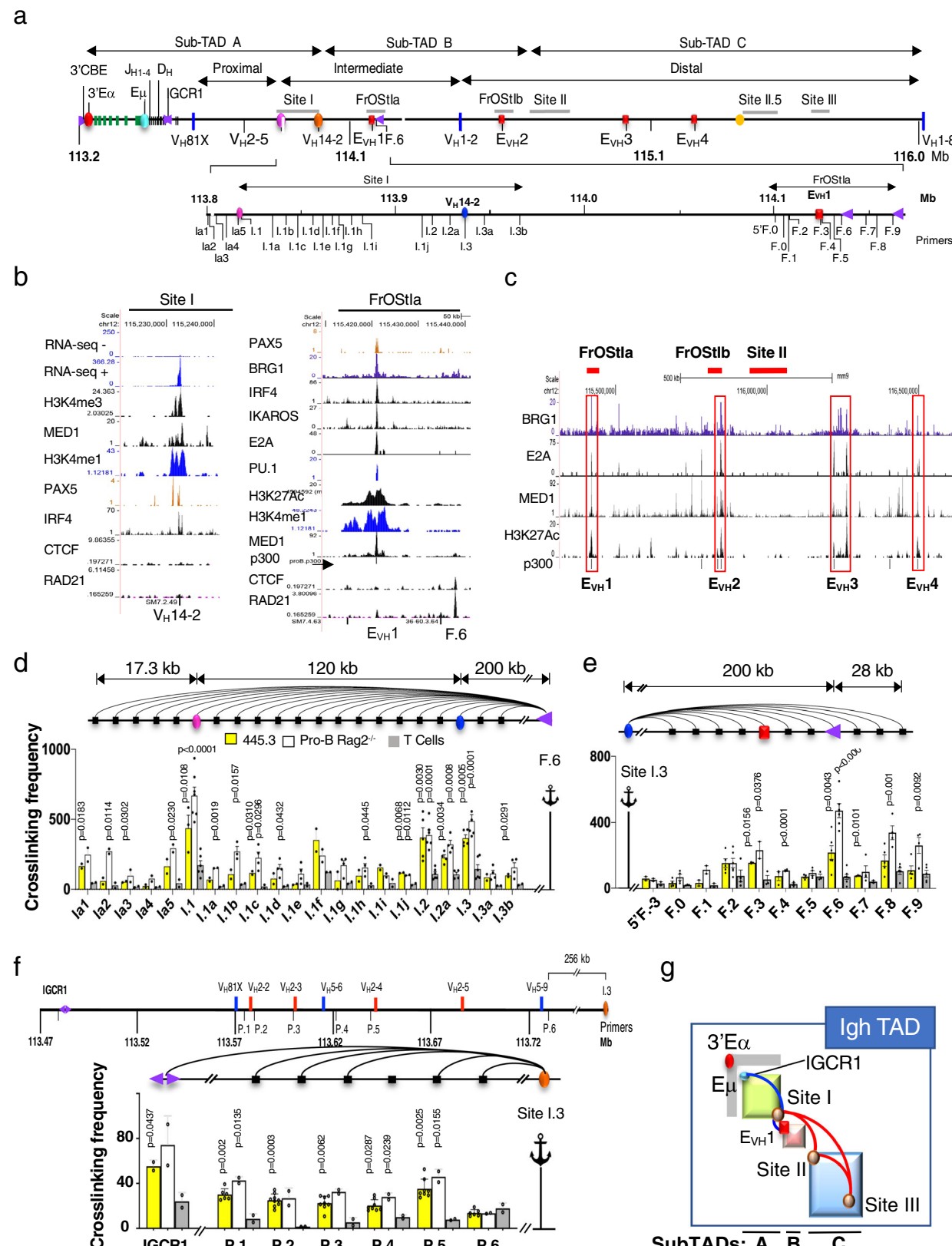

Eμ proximal recombination center (RC) that forms over the $J_H$ cluster and to which RAG1/2 is recruited[34]. Cells were stably complemented with a RAG1 expression construct and treated with STI-571 to induce endogenous RAG2 and VDJ recombination[35]. D->$J_H$ rearrangements, assessed in gDNA using the DFL.16 and $J_H$1 primers, were largely comparable in control and KO lines indicating that the first step in V(D)

J recombination is intact and independent of TE and CBE deletions (Fig. 2f, g). Slightly reduced D->$J_H$ recombination in $E_{VH}$1 KO.1 may be attributable to subclone variability (Fig. 2f, g).

$V_H$81X, is the most highly used $V_H$ gene in Abl-t pro-B cell lines[36] and in mouse C57Bl/6 pro-B cells[37] whereas distal $V_H$558->$DJ_H$ recombination is rare in Abl-t pro-B cell lines[36]. $V_H$7183->$DJ_H$

**Fig. 1 | Identification of Site I and FrOStIa loop anchors motifs. a** Schematic of the Igh locus with genomic coordinates (chr12, mm10) the directionality of which follows chromosome 12 showing the Igh sub-TADs A-C[17] and DNA elements (3'Eα, red dot; CBEs, purple arrows with orientation indicated; Eγ, teal dot; Eμ, pink square; $E_{VH}1$ (chr12:114182400-114183200), $E_{VH}2$ (chr12:114609100-114609900), $E_{VH}3$ (chr12:115023400-115024300), $E_{VH}4$ (chr12:115257500-115258400), red squares; $C_H$ region genes, green bars; $D_H$ and $J_H$ exons, black bars; $V_H$ gene segments, blue bars. The 3C primer sites in Site I and FrOStIa are indicated at bottom. **b** Public Chip-seq and RNA-seq positive (+) and negative (-) strand analyses for $V_H$14-2 and $E_{VH}1$ and F.6 CBE were generated from Rag deficient pro-B cells (Supplementary Table 11). **c** ChIP-seq studies identify $E_{VH}1$, $E_{VH}2$, $E_{VH}3$ and $E_{VH}4$ (red rectangles). **d**–**f** 3C assays. Arcs (3C assays), primers are identified below the graphs, 3C probes (anchor symbol). Average crosslinking frequencies are from

independent chromatin samples as indicated. Statistical comparisons are to T cells. $P$ values from two-tail Student's $t$ test and SEMs. Source data are provided as a Source Data file. **d** 3C assays analyzing Site I anchored at F.6 CBE. Chromatin samples for Abl-t 445.3.11 line, $n = 3$; $Rag2^{-/-}$ pro-B cells, $n = 3$; T cells: $n = 3$. **e** 3C assays analyzing FrOStIa and anchored at Site I.3. Chromatin samples for Abl-t 445.3.11 line, $n = 2$; $Rag2^{-/-}$ pro-B cells, $n = 2$; T cells, $n = 2$. **f** (*Upper panel*) Schematic of subTAD A showing a subset of $V_H$ genes and 3C primers (P) with genomic coordinates (chr12, mm10). (*Lower panel*) 3C assays using the Site I.3 (I.3) anchor and P.1-P.6 primers. Chromatin samples for Abl-t 445.3.11 line, $n = 6$; $Rag2^{-/-}$ pro-B cells ($n = 2$); T cells ($n = 2$). **g** Schematic of the Igh locus with looping interactions between IGCR1:Site I (subTAD A; blue and orange dots), Site I:$E_{VH}1$ (subTAD A-B; orange and red dots), and between Sites I:II, Sites II:III and Sites I:III (all subTADs; red arcs)[17].

rearrangements were analyzed in qRT-PCR using two forward primer sets specific for $V_H$81X (gray arrow), and a pan specific $V_H$7183 primer (red arrow) in combination with the reverse $J_H$1 primer (blue arrow) (Fig. 2f). Although $V_H$7183 rearrangement levels were strikingly reduced in all Abl-t pro-B cell lines harboring TE and CBE deletions, $V_H$81X rearrangements were significantly diminished only in $E_{VH}1$ KO lines (Fig. 2h). It is possible that the differences we detect for $V_H$81X and $V_H$7183 rearrangements in the KO lines are related to a sensitivity limitation of the assay due to the relatively low level of $V_H$7183 recombination and GLT expression. Therefore, we asked if $V_H$81X GLT expression was correlated with the propensity to engage in rearrangement. Indeed, $V_H$81X GLTs were undetectable in the $E_{VH}1$ KO lines consistent with the significant reduction in VDJ recombination but only partially diminished in $V_H$14-2 Pr- and F.6 CBE KO line suggesting that even low-level transcription may contribute to $V_H$81X- > $DJ_H$ recombination (Fig. 2e). Thus, $E_{VH}1$ deletion led to the deepest deficits for both transcription and recombination relative to other KOs (Fig. 2e, h).

## A Eμ-$V_H$Pr-$E_{VH}1$ chromatin hub mediates Igh locus architecture

Efficient $V_H$- > $DJ_H$ recombination, transcription and chromatin loop formation all require intact $V_H$14-2 Pr, $E_{VH}1$ and F.6 CBE motifs suggesting that these elements organize essential structure/function relationships in the Igh locus. $V_H$14-2Pr participates in a multiplicity of contacts in subTAD A including IGCR1 and with $E_{VH}1$ in subTAD B indicating a high degree of connectivity (Fig. 1e, f). To explore the role of $V_H$ Pr-E interactions we evaluated the overall topology of the locus spanning Eμ to $E_{VH}1$ and the formation of a regulatory hub using high resolution 3D DNA FISH and short probes (4.8-7.5 kb) for Eμ (green), 3'Eα/Site I.3 (3'Eα/$V_H$14-2 Pr; red) and $E_{VH}1$ (blue) (Fig. 3a). To explore the topology of the Igh locus we measured mean spatial distances[38] between FISH probes (Fig. 3a, b). The mean spatial distances separating the Eμ anchor probe from the 3'Eα and Site I.3 probes increased as a function of genomic distance and then leveled off between Site I.3 and $E_{VH}1$ in controls indicating locus compaction (Fig. 3c). In contrast, overall mean spatial distances increased in $E_{VH}1$ and F.6_CBE KO lines indicating substantial decompaction in subTAD A-B upon loss of these elements (Fig. 3c, d). The increased spatial distance separating Eμ and 3'Eα was confirmed in 3C studies which showed reduced crosslinking frequency for Eμ:3'Eα in the $E_{VH}1$- and F.6_CBE KO lines as compared to the control line (Fig. 3e). Deletion of the $V_H$14-2 Pr led to a dramatic increase of average spatial distance between Eμ and $E_{VH}1$ whereas Eμ and Site I.3 were more compacted relative to the control in FISH assays (Fig. 3c, d). Thus, Eμ:Site I.3 association is not mediated by direct interaction with the $V_H$14-2 Pr and must therefore be supported by another element. We conclude that $E_{VH}1$ and F.6_CBE have a profound influence on locus compaction in subTADs A-B and that the $V_H$14-2 Pr:$E_{VH}1$ interaction serves to bridge $E_{VH}1$ to Eμ possibly through its interaction with IGCR1 (Fig. 1f).

Molecular contacts between chromatin elements are indicated by superimposed FISH probes (≤0.3 μM). Segregation of probe contacts into structural categories allows identification of

pure pairwise and three-way overlap interaction frequencies in single cells. Importantly, pure pairwise Site I.3- $E_{VH}1$ interactions were significantly diminished in all three KO lines in accord with our 3C analyses (Figs. 2c, 3f). Probes for Eμ, Site I.3 and $E_{VH}1$ were superimposed or overlapped as beads on a string in varying order in ~20% of control alleles whereas the frequency of these probe configurations trended lower- or were significantly reduced in all three KO lines (Fig. 3f, inset a-d). Thus, formation of a chromatin hub requires the presence of the $V_H$14-2 Pr-, $E_{VH}1$- and F.6 CBE. Significantly, the frequency of distantly spaced probes (≥0.31 μM) in which no overlap occurs greatly increased from ~20% of alleles in controls to 40-50% in the KO lines demonstrating an overall destabilization of subTAD A-B conformation (Fig. 3f inset h-i). The reduced presence of three-way interactions and the increased topological instability are consistent with reduced V- > DJ recombination found for these KO lines. Eμ-Site I.3 looping increased upon $V_H$14-2 Pr KO suggesting that this Pr blocks these interactions, perhaps by binding with alternative interaction targets (Fig. 3f inset e). The Eμ- $E_{VH}1$ configuration was largely impervious to TE deletion indicating that this pairwise conformation may be formed through other interactions (Fig. 3c). An ensemble of topological configurations detected in single cells including the Eμ-$V_H$14-2 $E_{VH}1$/F.6 CBE chromatin hub are schematically represented and together represent the population average locus conformation (Fig. 3g). Hence, the $V_H$14-2 Pr, $E_{VH}1$ and F.6_CBE have a profound influence on 3'Igh locus compaction and the spatial accessibility of proximal $V_H$ genes to the RC.

## $V_H$ gene usage is altered in mice lacking $E_{VH}1$

To assess the influence of $E_{VH}1$ on Igh repertoire and locus conformation in a physiological setting, we deleted 515 bp spanning $E_{VH}1$ using a genome editing strategy in mice (Fig. 4a). To analyze the pre-selected Igh repertoire we FACS purified pro-B cells from WT and $E_{VH}1^{-/-}$ mice, and performed VDJ-seq, an unbiased assessment method for $V_H$ segment usage in $VDJ_H$ junctions in gDNA (Supplementary Fig. 4a)[39]. The overall profile for $V_H$ gene usage in our WT samples compares well to previously reported findings[39] (Fig. 4b). In contrast, deletion of the $E_{VH}1$ element led to significantly reduced $V_H$ gene usage, in a 335 kb domain which we refer to as the $E_{VH}1$ zone of influence (ZOI), extending from 5' of $V_H$81X ($V_H$7183.2.3) to the $V_H$J606 genes that mark the end of the intermediate genes and are located approximately halfway through subTAD B (Fig. 4c, d). $V_H$ genes closest to $E_{VH}1$ are more severely reduced, in general, by the absence of $E_{VH}1$. $V_H$ gene usage outside the ZOI was sporadically altered with increased usage near $E_{VH}2$ and $E_{VH}3$ implying widespread structure/function changes in $E_{VH}1^{-/-}$ alleles (Fig. 4b, c).

We observe that $V_H$81X usage is not diminished in the VDJ-seq analysis of $E_{VH}1^{-/-}$ pro-B cells in contrast to what was observed in the $E_{VH}1$ KO Abl-t lines (Fig. 2h). It should be noted that the repertoire of rearrangements is very limited in Abl-t cell lines, with $V_H$81X being the dominant $V_H$ gene used for rearrangement[36],

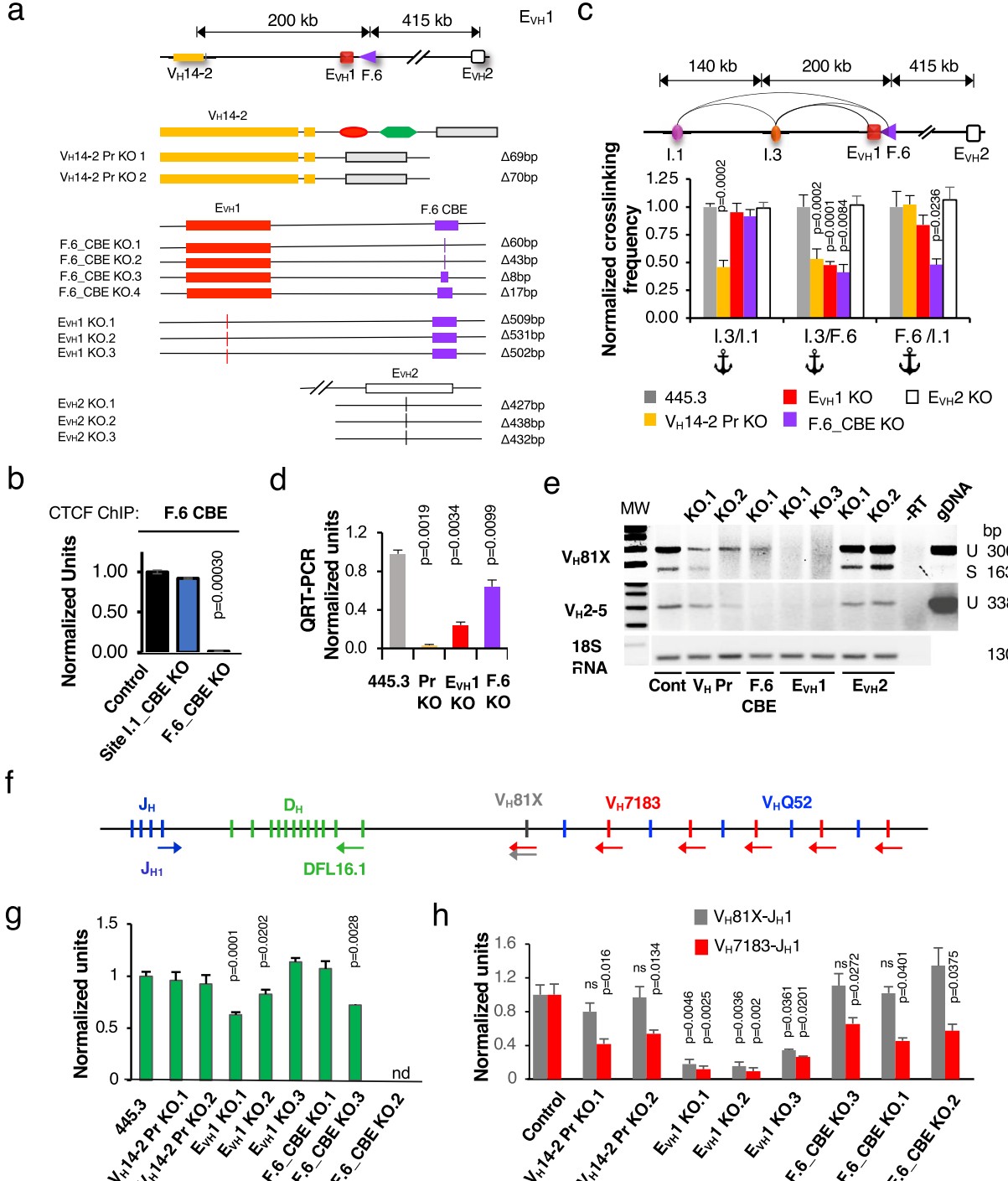

**Fig. 2 | Igh locus function is dependent on a V_H promoter, an E_VH and CBE in Abl-t pro-B cells.** Source data are provided as a Source Data file. All p values are from two-tail Student's t test. **a** (*Upper panel*) The Igh locus from the V_H14-2 gene to E_VH2. (*Lower panel*) Biallelic identical deletions constructed for the V_H14-2 promoter (V_H14-2 Pr, yellow rectangle), F.6 CBE (F.6_CBE, purple triangle), E_VH1 (red square), and E_VH2 (white square) in the Abl-t 445.3.11 cell line. **b** ChIP assays for CTCF binding at the F.6 CBE in the control (n = 3), Site I.1- (n = 3) and F.6 CBE (n = 3) KO lines. **c** 3C looping assays. (*Upper panel*) Map of genomic distances. (*Lower panel*) Normalized 3C crosslinking frequencies from four experiments in which the 455.3.11 (n = 10) or 445.3.8 (n = 2) matched controls were set to 1.0 and represented once. The crosslinking frequencies were averaged for chromatins from each KO type (E_VH1 KO, n = 9; E_VH2 KO, n = 6; F.6_CBE KO, n = 6; F.6_CBE KO, n = 8; VH14-2 Pr KO (n = 6) and normalized to the control. **d** QRT-PCR assays for the V_H14-2 gene using 18 S RNA as a loading control. Samples were from control (n = 14) VH14-2 Pr KO

(n = 8), E_VH1 KO (n = 6), F.6_CBE KO (n = 9). **e** RT-PCR of V_H81X and V_H2-5 GLTs and the 18 S RNA loading control were harvested at 32 and 28 cycles, respectively. Cells were STI-571 (2.5 mM) treated for 48 h. **f** Schematic of the D_HJ_H clusters and proximal V_H gene segments (vertical bars), with primer positions (colored arrows) used to analyze D-J and V-DJ rearrangements. **g, h** Lines were stably complemented with RAG1 and STI-517 treated. **g** D-J rearrangements in gDNA were analyzed by qPCR with DFL16.1 and J_H1 primers. Samples were from 455.3.11 (n = 4), 455.3.8 (n = 5), V_H14-2 Pr KO (n = 4), E_VH1 KO (n = 6), F.6_CBE KO (n = 4). **h** Normalized qRT-PCR data for V- > DJ rearrangement using V_H81X and V_H7183 primers in combination with J_H1. V_H81X-J_H1 analyses were from control 455.3.11 (n = 4), 455.3.8 (n = 3), E_VH1 KO (n = 6), V_H14-2 Pr KO (n = 3), F.6_CBE KO (n = 4). Samples numbers for VH7183-JH1 analyses were identical to that of V_H81X-JH1 except for 455.3.11 (n = 12), V_H14-2 Pr KO.1 (n = 5) and F.6_CBE KO.2 (n = 4).

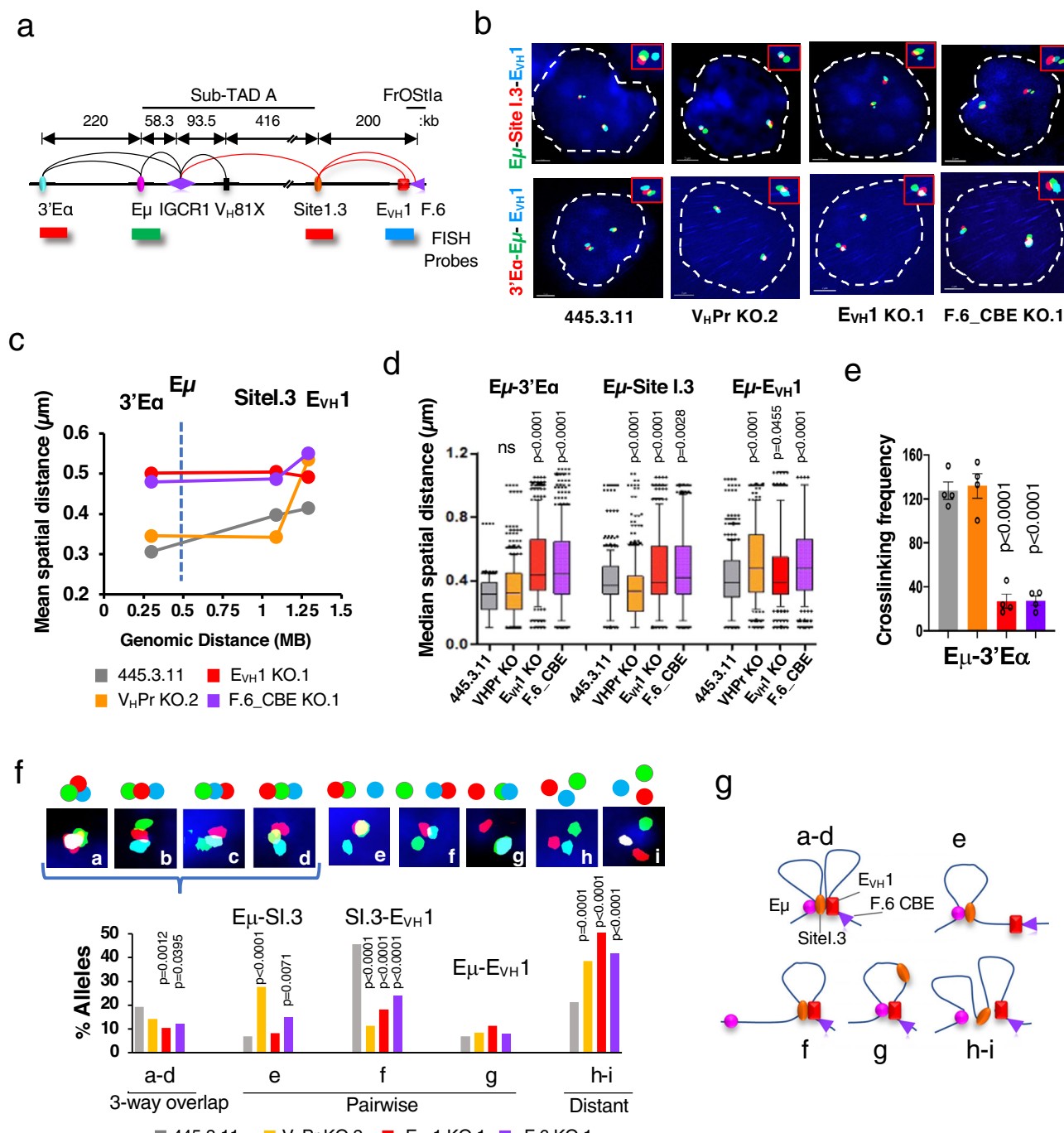

**Fig. 3 | Transcriptional elements regulate locus compaction in Abl-t pro-B cells.** Source data are provided as a Source Data file. **a** The Igh locus from 3'Eα to FrOStIa and the position of FISH probes. Chromatin loops detected in 3C studies from pro-B cells (red arcs; Fig. 1d, f) and previous work (black arcs)[4,57,78]. FISH probes are separated by 220 kb (Eµ-3'Eα), 568 kb (Eµ-Site I.3) and 758 kb (Eµ-E_VH1). **b** Representative nuclei from fixed 445.3.11 and KO lines were simultaneously hybridized in duplicate with the indicated probes (y-axis). Short probes were labeled with AlexaFluor 488 (Eµ, green), AlexaFluor 555 (Site I.3 or 3'Eα, red), and AlexaFluor 647 (E_VH1, blue). Scale = 2 µm. **c** Mean spatial distances (y-axis) were plotted as a function of genomic distance (x-axis) with lines indicating connectivity only, for 404 alleles from two independent experiments for each probe combination[27]. Vertical dashed line shows the Eµ anchor. **d** N = 404 alleles/genotype over two independent experiments. Box plots represent the distribution of spatial distances with medians (middle line), 25th and 75th percentile (box), and 10th

and 90th percentile (whiskers), and outliers (single points) shown. P values from two-tailed nonparametric Mann-Whitney U test. **e** 3C crosslinking frequencies for Eµ:3'Eα looping interactions anchored at Eµ, analyzed in chromatin samples from the 445.3.11 control (n = 4) and KO (n = 4) lines except for E_VH1 KO (n = 7) with SEMs. P values from unpaired two tailed Student t test. **f** Quantitation of 3D probe configurations in three-color DNA FISH. Pairwise and three-way distances between the red, green, and blue DNA FISH signals were divided into three categories: <0.3, 0.31–0.5, >0.5 µm. Nine probe configurations (upper panel) and their frequencies (lower panel) are shown. Inset a: Three-way overlap with all probes <0.3 µm from each other, insets b–d: beads on a string configuration with probes 1-2 and 2-3 or 1–3 < 0.3 µm from each other, insets e–g: pairwise probe configurations with two probes <0.3 µm from each other, inset h: all probes distanced by 0.31–0.5 µm, inset i: all probes distanced by >0.5 µm. P values, two tailed Chi square tests. **g** Representation of Igh allele configurations.

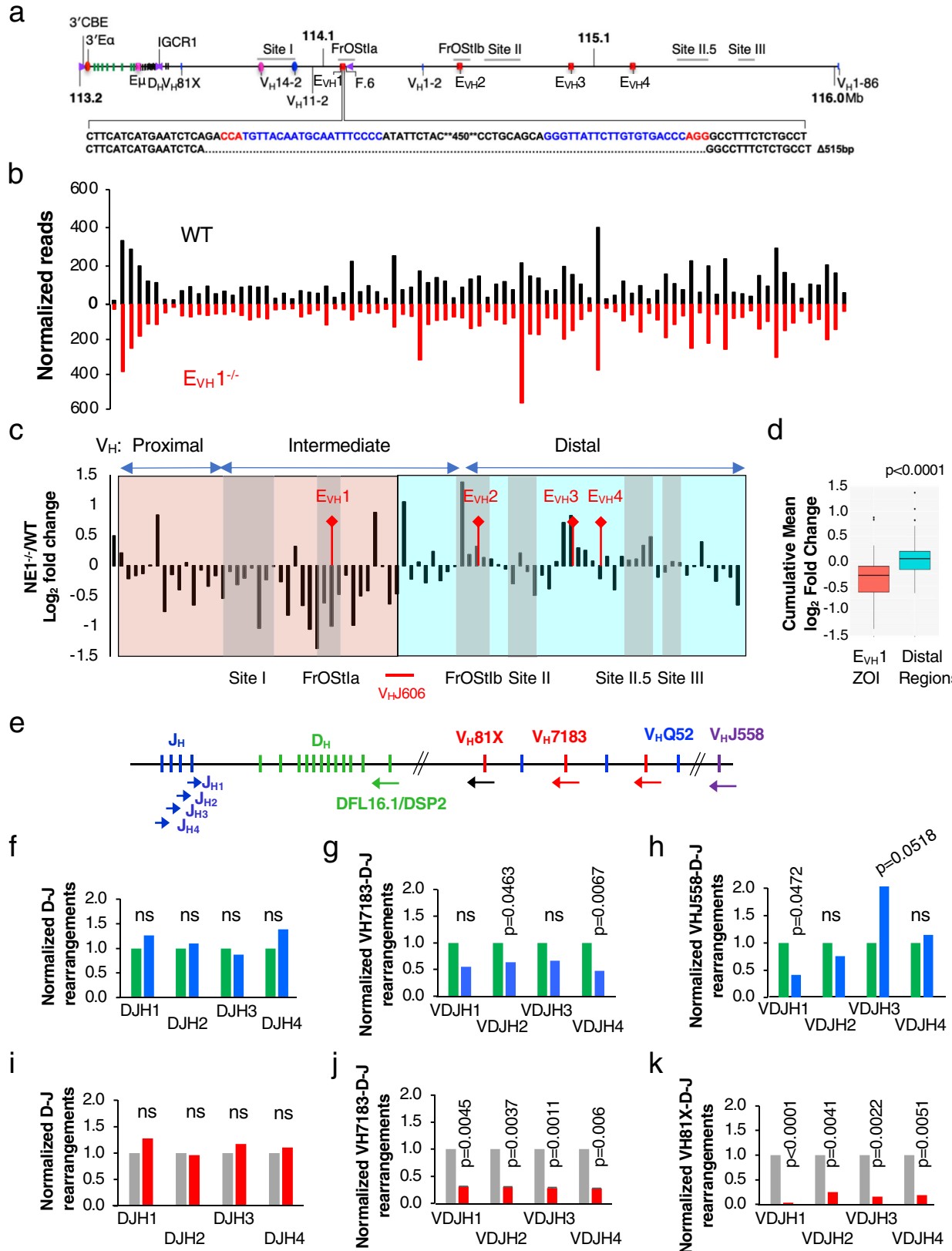

presumably due to the absence of locus contraction[9] so that the repertoires are quite different in these two cell types. More relevant to this issue, VDJ-seq in pro-B cells allows analysis of the relative ratio of $V_H$ gene usage within normalized data sets but cannot determine if the absolute level of rearrangement is lower in one sample compared to another. Therefore, to address the

possibility that the overall frequency of proximal $V_H$ gene rearrangements was lower in the $E_{VH}1^{-/-}$ pro-B cells despite the relative usage of $V_H7183$ genes being similar, we used pan-specific $V_H7183$ and $V_HJ558$ primers with reverse primers for each $J_H$ segment[40] and analyzed VDJ rearrangements in gDNA of WT and $E_{VH}1$ KO cells in both primary pro-B cells and the Abl-t lines (Supplementary

**Fig. 4 | $E_{VH}1$ regulates regional $V_H$ gene usage in mice.** Source data are provided as a Source Data file. **a** $E_{VH}1$ was deleted in mice using genome editing. Guide RNAs (blue) with PAM motifs (red) are indicated with the $E_{VH}1$ deletion. **b, f–k** $P$ values from unpaired two-tailed Student's $t$ test. **b** VDJ-seq analysis of gDNA from WT and $E_{VH}1^{-/-}$ pro-B cells. WT ($n = 2$), $E_{VH}1^{-/-}$ ($n = 2$). $V_H$ gene reads from two independent experiments were normalized. **c** Ratio of $E_{VH}1^{-/-}$/WT normalized reads. Chromatin loop anchors (grey vertical bars) and $V_H$J606 genes (red line) are indicated. **d** VDJ-seq cumulative fold change ($E_{VH}1^{-/-}$/WT) in ZOI (red) and distal (blue) regions **(c)** using the two tailed Mann Whitney Wilcoxon test for p values. Box plots show the mean (middle line), 25th and 75th percentile (box), 10th and 90th percentile (whiskers) and outliers (single points). **e** Schematic of the $D_HJ_H$ clusters and proximal $V_H$ gene segments (vertical bars), with $J_H$ and pan-specific $V_H$7183 primer positions (colored arrows) used to analyze D-J and V-DJ rearrangements in qPCR assays as described[40]. **f–h** Genomic DNA was prepared from primary pro-B cells isolated by FACS from WT (green bars) and $E_{VH}1^{-/-}$ (blue bars) mice (Supplementary Fig. 4b). WT ($n = 2$), $E_{VH}1^{-/-}$ ($n = 2$). Each sample was pooled from three mice. **f** D-J rearrangements were analyzed with the degenerate DFL16.1/DSP2 primer in combination with each of four $J_H$ primers and normalized values were set to 1.0 for WT. **g, h** V-DJ rearrangements were amplified using the $V_H$7183 (**g**) or the $V_H$J558 (**h**) primer in combination with each of four $J_H$ primers and normalized values were set to 1.0 for WT. **i–k** GDNA were prepared from control ($n = 2$) (gray bars) and $E_{VH}1$ KO.1 and KO.2 ($n = 2$) lines (red bars). **i, j, k** D-J (**i**), $V_H$7183-DJ (**j**), and $V_H$81X-DJ (**k**) were analyzed in combination with each of four $J_H$ primers and normalized values were set to 1.0 for the control. $P$ values from unpaired two-tailed Students $t$ test.

Fig. 4b) (Fig. 4f–k). $V_H$7183 rearrangements in conjunction with each $J_H$ segment are significantly diminished or trended lower in $E_{VH}1^{-/-}$ pro-B cells as compared to WT whereas D-J recombination is unaffected (Fig. 4f, g). Even more pronounced, $V_H$7183 and additionally the $V_H$81X gene rearrangements are greatly reduced in the Abl-t $E_{VH}1$ KO line whereas D-J recombination is intact in accord with our findings (Fig. 2g, h) (Fig. 4i, j, k). Hence, there is an overall similarity in the impact of $E_{VH}1$ deletion on $V_H$7183 gene usage in pro-B cells and Abl-t pro-B lines. In contrast, there are variable effects of $E_{VH}1$ deletion on $V_H$J558 rearrangements with different $J_H$ segments in pro-B cells which may reflect the variable impact of $E_{VH}1$ deletion on $V_H$J558 usage found in the VDJ-seq study (Fig. 4c, h). We conclude that the frequency of V(D)J rearrangement is reduced for the proximal $V_H$ genes in the $E_{VH}1^{-/-}$ mice. Collectively, these findings indicate a consistent diminution of VH7183 usage upon loss of $E_{VH}1$.

## Splenic B1a cells binding Ptc are diminished in $E_{VH}1^{-/-}$ mice

The $E_{VH}1$ ZOI encompasses the $D_H$ proximal $V_H5$ ($V_H7183$) and $V_H2$ ($V_H$Q52) families and the small intermediate gene families including the $V_H11$ and $V_H12$ exons that are frequently expressed in B1 B cells[41,42]. $V_H11.2$ in particular is highly over-represented in B1a B cells, and such antibodies are reactive with PtC[41]. Because $V_H11.2$ rearrangement was reduced 2-fold in $E_{VH}1^{-/-}$ as compared to WT pro-B cells (Fig. 4b, c), we examined the presence of B cell compartments and the frequency of B1a- and PtC binding cells. $E_{VH}1$ deletion had little effect on B cell development as comparable proportions of Hardy fractions (B-C', pro-B; D, pre-B; E, immature B; F, mature B) in the BM and marginal zone and follicular B cells in the spleen are detected in WT and $E_{VH}1^{-/-}$ mice (Supplementary Fig. 5a–g). Although normal proportions of peritoneal B1 B cells were detected in $E_{VH}1^{-/-}$ (Supplementary Fig. 5f, g), splenic B1 B cells expressing surface $CD19^+B220^{lo}CD23^-IgM^{hi}IgD^{lo}$ were significantly reduced in $E_{VH}1^{-/-}$ relative to WT mice (Fig. 5a, b). Moreover, B1a cells binding PtC are depleted in $E_{VH}1^{-/-}$ as compared to WT mice (Fig. 5c, d). The reduced rearrangement of $V_H11.2$ in the $E_{VH}1^{-/-}$ pro-B cells is therefore a possible explanation for the reduction of splenic B1a cells and those binding PtC. Thus, perturbation of the preselected Igh repertoire in pro-B cells has potential ramifications for the peripheral repertoire.

## $E_{VH}1$ is an Igh locus architectural element in mice

To define the contribution of $E_{VH}1$ to locus architecture, in situ Hi-C data sets[14] were generated and biological replicates were merged (Supplementary Fig. 6). Rag deficient pro-B cells are used to ensure that the Igh locus has not undergone V(D)J recombination and are compared with those from mouse embryonic fibroblasts (MEF) cells in which the Igh locus is inactive[43]. Pooled Hi-C samples recapitulated reported chromatin structures with high reproducibility scores and similar data quality (Supplementary Table 10) (Supplementary Fig. 6a)[14]. The Igh 2.9 Mb TAD (yellow box) in pro-B cells displays strong locus boundaries (green/blue arrows) and rarely interacts with other compartments (black brackets) as previously observed

(Supplementary Fig. 6a, b)[17]. In contrast, the 5' Igh TAD boundary in MEF is subsumed into an adjacent TAD and interacts with other compartments indicating a B cell specific difference in TAD structure (Supplementary Fig. 6a, b).

To identify pro-B cell specific interactions we created Hi-C difference heatmaps by subtracting MEF- from pro-B- derived contacts, a strategy used in our earlier studies[17] and adapted to Hi-C data sets (Methods). The $Rag1^{-/-}$ minus MEF difference map reveals nested self-interacting domains (vertical dashed lines) spanning hundreds of kb along the locus that have been schematically summarized (Fig. 6a, bi, iii). We detect stripes (blue arrowheads) originating from the 3'RR and IGCR1 as previously observed[23,44] that are absent in MEF (Fig. 6bi) and are also visible in the unsubtracted Hi-C data (black arrows) (Supplementary Fig. 6b, c). Architectural stripes reflect interactions between a single anchor and a continuum of genomic elements, are considered evidence of cohesin mediated loop extrusion[19,44] which likely facilitates V(D)J joining in pro-B cells[9]. These observations support a major role of IGCR1 as a downstream anchor of loop extrusion for the $V_H$ locus rather than Eμ as previously suggested[9]. We observe a series of stripes (black arrowheads) extending in a 5'->3' direction from the corners of multiple self-interacting domains (Fig. 6bi). Several of these stripes meet off the diagonal to form "dots" indicating the presence of IGCR1-$E_{VH}1$ and IGCR1-$E_{VH}2$ (marked a and b), $E_{VH}1$-$E_{VH}2$ (marked c), and $V_H$J606 genes-$E_{VH}2$ (marked d) anchored loops indicating that $E_{VH}$-$E_{VH}$ and $E_{VH}$-Pr contacts contribute to Igh locus conformation (Fig. 6biii). Corner dots are most frequently anchored by CBE in a convergent orientation, bound by the CTCF architectural protein and represent persistent loops[14,45]. All CBEs bound by CTCF in the $V_H$ domain are convergent with the IGCR1 CBEs[23]. Each of the $E_{VH}$s is paired with a nearby CBE bound by CTCF and the cohesin component, RAD21 (Fig. 1b) (Supplementary Fig. 1). The F.6 CBE flanking $E_{VH}1$ interacts with Site I.3, and Site I.I and these contacts were abolished upon F.6_CBE deletion in Abl-t pro-B lines (Fig. 2c) (Supplementary Fig. 2c, d). These observations together suggest that the Hi-C dot domains were formed by loop extrusion that is impeded by CTCF/cohesion as noted[19,21,46]. Another possibility is that active enhancers, such as $E_{VH}1$s can function as barriers to loop extrusion[47,48].

The structural profile of the Igh locus was substantially altered in $Rag1^{-/-}E_{VH}1^{-/-}$ derived Hi-C difference maps (Fig. 6bi, ii) (Supplementary Fig. 7a, bi, ci). Many loops, represented by corner dots, involving $E_{VH}1$ were abolished including those between IGCR1-$E_{VH}1$ (marked a, grey) and $E_{VH}1$-$E_{VH}2$ (marked c, grey), whereas those involving the $E_{VH}2$ anchor such as IGCR1-$E_{VH}2$ (marked b), and $V_H$J606 genes-$E_{VH}2$ (marked d) are retained in the $Rag1^{-/-}E_{VH}1^{-/-}$ pro-B cells (Fig. 6bii, iv) (Supplementary Fig. 7bi, ci, d). Deletion of $E_{VH}1$ also led to the new appearance of IGCR1-Site I (marked e, purple circle) interactions, the formation of the δ loop (dashed box) configured by the association of Site I-$V_H$J606 genes and a series of stripes and chevrons (purple arrowheads) (Fig. 6bii, iv) (Supplementary Fig. 7bi, ci, d). Active enhancers can function as TAD boundary elements and spatial proximity with promoters can be achieved during the extrusion process[47,48]. In this context, it is evident that $E_{VH}1$ (green dot) directly participates in

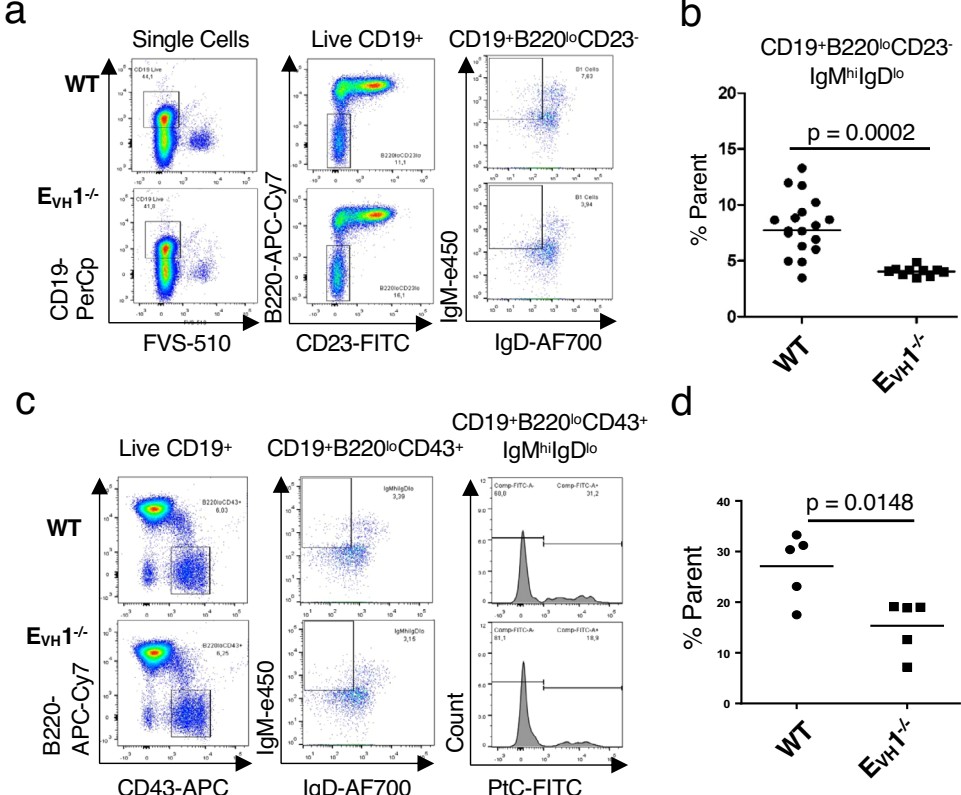

**Fig. 5 | Phosphatidylcholine binding splenic B1a cells are reduced in $E_{VH}1^{-/-}$ mice. a** Representative flow cytometry plots of splenocytes gated for B1 B cells were assessed using FVS-510 and antibodies (CD19-PerCp, B220-APC Cy7, CD23-FITC, IgM-e450, IgD-AF700). **b, d** Frequencies of B1 B cells were averaged and each symbol represents results from one mouse. **b** Average frequency of B1 B cells (CD19⁺B220$^{lo}$CD23$^-$IgM$^{hi}$IgD$^{lo}$) was assessed in three independent experiments. WT ($n = 35$), $E_{VH}1^{-/-}$ ($n = 19$) mice. $P$ values from unpaired two-tailed Student's $t$ test. **c** Representative flow cytometry plots of splenocytes gated for B1 B cells using FVS-510 and antibodies (CD19-PerCp, B220-APC Cy7, CD43-APC, IgM-e450, IgD AF700) and PtC liposome (FITC). **d** Average frequency of B1 B cells binding PtC (CD19⁺B220$^{lo}$CD43⁺IgM$^{hi}$IgD$^{lo}$PtC⁺). WT ($n = 5$), $E_{VH}1^{-/-}$ ($n = 5$) mice. $P$ values from unpaired two-tailed Student's $t$ test.

loop extrusion as stripes emanating from this position and moving toward IGCR1 disappear from the Hi-C difference map in $Rag1^{-/-}E_{VH}1^{-/-}$ pro-B cells (Fig. 6bi, ii). Notably, impairment of Igh locus structure conferred by $E_{VH}1^{-/-}$ extends from IGCR1 to the J606 genes and maps to the $E_{VH}1$ ZOI, thus, linking Igh spatial architecture with $V_H$ gene utilization.

**Loss of $E_{VH}1$ leads to locus hypercontraction**

To explore the contribution of $E_{VH}1$ to the higher order locus conformation we compared the spatial distances between BAC FISH probes H14, Eμ-IGCR1 (referred to as Eμ) and the RI, RII, and RIII probes, corresponding to Sites I, II and III, hybridized in $Rag2^{-/-}$ and $Rag2^{-/-}E_{VH}1^{-/-}$ pro-B cells (Fig. 6c, d)[17]. We used probe RI as a reference and determined spatial distances with all other probes. Although spread over a large genomic region the average spatial distance separating probes RI from Eμ, RII, and RIII were similar whereas spatial distances to the H14 probe outside the locus were large indicating that most $V_H$ genes are at comparable distances to the RC in $Rag2^{-/-}$ pro-B cells, as previously observed (Fig. 6e, f)[27]. In contrast, closer spatial distances are detected between the Eμ and RI probes with RIII in $Rag2^{-/-}E_{VH}1^{-/-}$ pro-B cells reflecting hyper-compaction with the distal $V_H$ gene segments relative to the control (Fig. 6e, f). Likewise, closer spatial distances are detected between the RI anchor with RII and RIII probes and greater distance to H14 in the Abl-t $E_{VH}1$ KO line reflecting increased locus compaction in the distal regions of the Igh locus relative to the control (Supplementary Fig. 8). Thus, $E_{VH}1$ deletion is correlated with increased overall locus compaction arguing for a link between long-distance loop extrusion and locus compaction.

**$E_{VH}1$ coordinates an enhancer network**

To explore the scope of $E_{VH}$ interactions and their impact on Igh locus topology we reduced Hi-C dimensionality by deriving virtual 4C datasets. Using $E_{VH}1$, Eμ, and IGCR1 viewpoints we identified high-frequency interactions defined as in the top 15 (star) or 25 (circle) percent of all contacts in both biological replicates for $Rag1^{-/-}$ (black symbols) and $Rag1^{-/-}E_{VH}1^{-/-}$ (red symbols) pro-B cells (Fig. 7a). $E_{VH}1$ was highly interactive with its own flanking region, Eμ, $E_{VH}2$, $E_{VH}3$, and $E_{VH}4$ in $Rag1^{-/-}$ pro-B cells (Fig. 7a). Although $E_{VH}3$ associates with $E_{VH}1$, its interaction frequency did not pass the threshold for high-frequency contacts in both replicates (Fig. 6A). Deletion of $E_{VH}1$ (red dashed line) led to loss of contact with all other $E_{VH}$s and with Eμ and the emergence of two new peaks (red arrows) of contact in $Rag1^{-/-}E_{VH}1^{-/-}$ pro-B cells and implies that $E_{VH}1$ coordinates an enhancer network (Fig. 7a).

Examination of the Eμ viewpoint confirmed interaction with 3'Eα hs4, Eγ, and IGCR1 as previously reported[3,4,9,49] and with $E_{VH}1$ (Fig. 6A). Eμ:$E_{VH}1$ contacts were apparently lost and new or increased Eμ interactions (red arrows) with Eγ, Site I.3, and $E_{VH}2$ were detected in $Rag1^{-/-}E_{VH}1^{-/-}$ pro-B cells suggesting that $E_{VH}1$ competes with Eγ and $E_{VH}2$ for interaction with Eμ (Fig. 7a). Finally, analysis of the IGCR1 viewpoint revealed interactions with 3'Eα hs4, Eμ, $V_H81X$ and $E_{VH}1$ but not with Eγ or with other $E_{VH}$s indicating that IGCR1 could bridge Eμ to $E_{VH}1$ (Fig. 7a). We infer that $E_{VH}1$ may integrate contacts with Eμ, IGCR1 and the distal $E_{VH}$s.

To independently confirm that Eμ interactions shift from $E_{VH}1$ to $E_{VH}2$ upon $E_{VH}1$ deletion we performed FISH and examined inter-probe distance distributions using short probes for Eμ (green), $E_{VH}1$ (blue), and $E_{VH}2$ (red) in $Rag2^{-/-}$ and $Rag2^{-/-}E_{VH}1^{-/-}$ pro-B cells and segregated

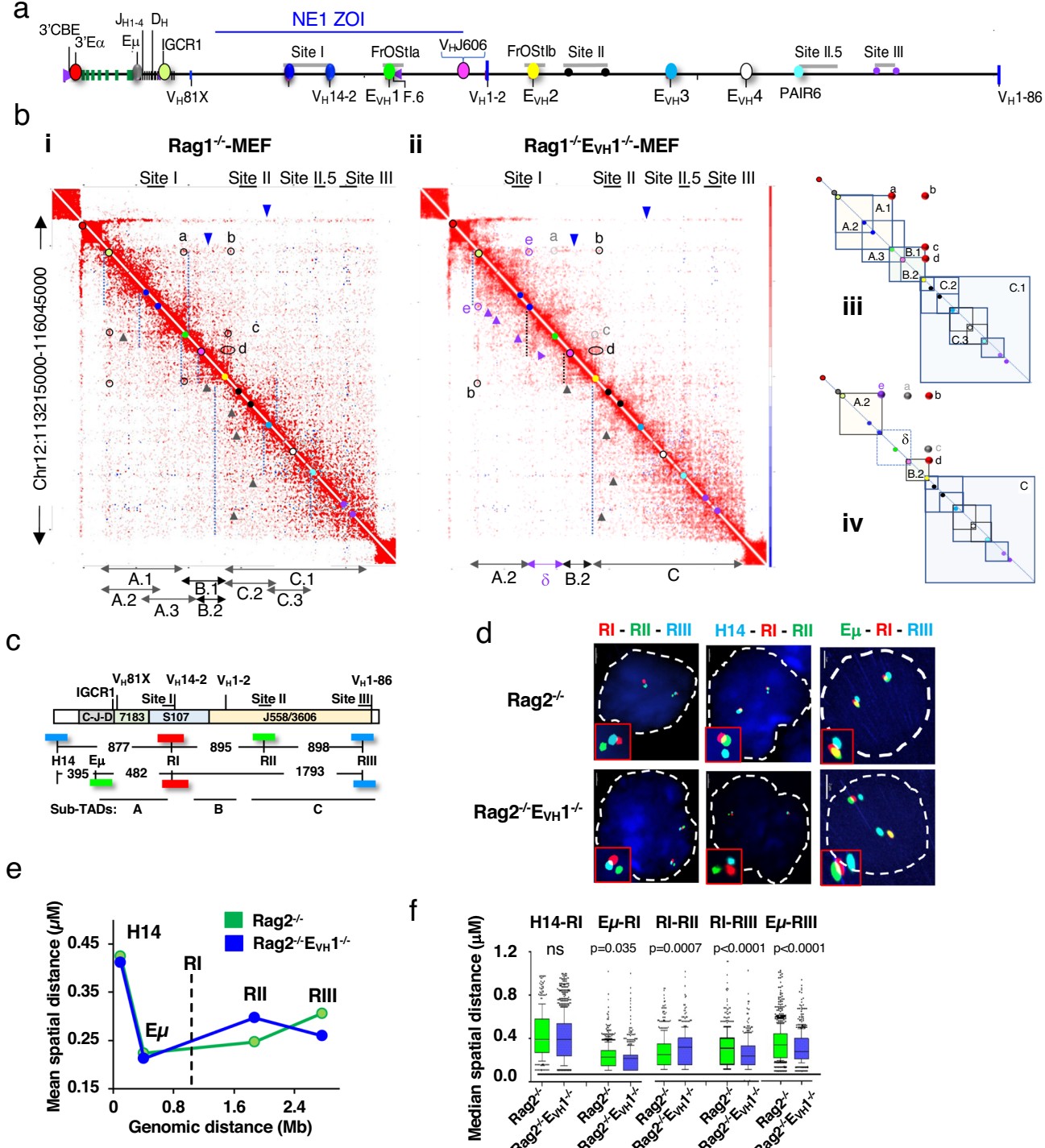

**Fig. 6 | E_VH1 modulates Igh locus topology and compaction in pro-B cells.**
**a** Schematic of the Igh locus. Eµ is situated between J_H and C_H genes and 3'Eα. The J_H and D_H segments are located between the constant region genes (C_H) and V_H7183 (V_H81X) segments. **b** Regulatory elements (colored dots) identified in **a** arrayed along the Hi-C diagonal. Hi-C difference heatmaps at 10 kb resolution from purified CD19⁺ pro-B cells from Rag1⁻/⁻ (n = 2) **(i)** and Rag1⁻/⁻E_VH1⁻/⁻ (n = 2) **(ii)** mice were generated (see Methods). Nested loop domains (dashed vertical lines), stripes (3'->5', blue triangles; 5'->3' black triangles), dots (black circles), lost dots (gray circles). Rag1⁻/⁻E_VH1⁻/⁻ specific stripes (purple triangles) and dots (purple circles). **iii, iv)** Schematic of nested loop domains in Hi-C difference heatmaps from Rag1⁻/⁻ **(iii)**, and Rag1⁻/⁻E_VH1⁻/⁻ **(iv)** pro-B cells. **c** The Igh locus includes the interspersed V_H J558/3609 (V_H1-86) families at the 5' end, the V_H7183 (V_H81X) family at the 3' end and the intermediate V_H segments (V_H14-2) include the S107 family with nine smaller V_H families. Representative V_H genes are positioned above the locus

(chr12, mm10), V_H81X (113578506 − 113578799), V_H14-2 (113994469-113994762), V_H1-2 (114469075-114469371) V_H1-86 (116000028-1160000321). BAC probes, H14, Eµ, RI, RII, RIII are indicated by their colors and are shown with the distances between them. **d−f** Source data are provided as a Source Data file. **d** Representative nuclei from fixed Rag2⁻/⁻ and Rag2⁻/⁻E_VH1⁻/⁻ pro-B cells hybridized with three labeled probe combinations in two biologically independent experiments. Scale bar, 2 µm. Rag2⁻/⁻ pro-B alleles for each probe pair: n = 430 (H14-RI), n = 541 (RI-RII), n = 922 (Eµ-RI, RI-RIII, Eµ-RIII). Rag2⁻/⁻E_VH1⁻/⁻ pro-B alleles for probe pairs: n = 1213 (H14-RI), n = 400 (RI-RII), n = 566 (Eµ-RI, RI-RIII, Eµ-RIII). **e** Mean inter-probe distances as a function of genomic distance. Lines indicate connectivity only. The RI anchor probe position, vertical dashed line. **f** Box plots show the median (middle line), 25th and 75th percentile (box) and 10th and 90th percentile (whiskers) and outliers (single points) of spatial distances between probes and anchor R1. P values from unpaired two-tailed nonparametric Mann–Whitney U test.

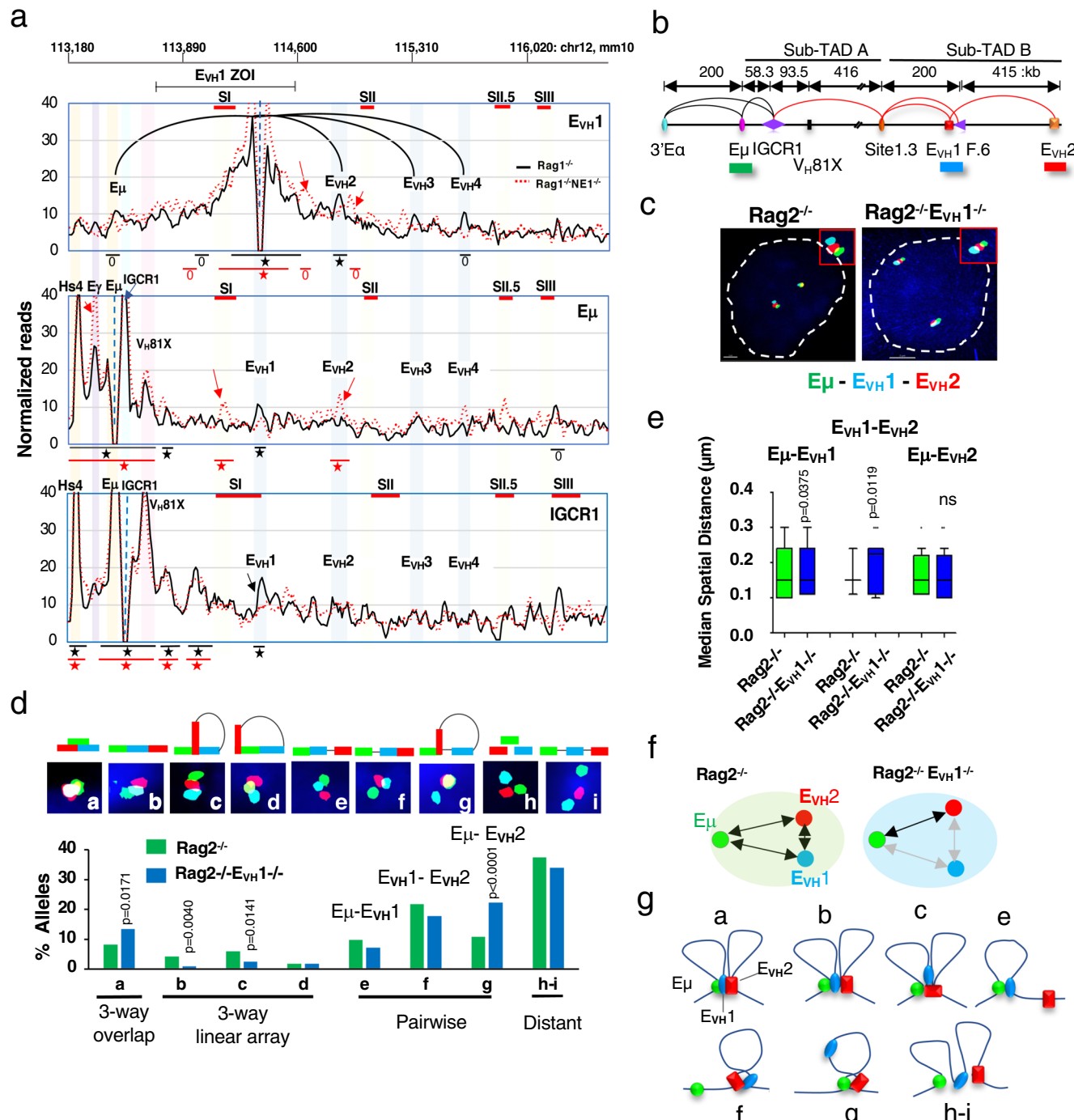

**Fig. 7 | E_VH interactome identified.** a Virtual 4C interactions were extracted from KR normalized Hi-C data sets from Rag1⁻/⁻ (black lines) and Rag1⁻/⁻E_VH1⁻/⁻ (dashed red lines) CD19⁺ pro-B cells. *Top*: Genomic coordinates (mm10) with the E_VH1 ZOI and Sites (S) I, II, II.5 and III. 4C viewpoints (dashed vertical line) in a 30 kb running window analysis with 10 kb steps from merged biological replicates with E_VH1 interactions (black arcs) shown. Stars (top 15%), circles (top 25%) of locus-wide interactions. New peaks (red arrows) and lost peaks (black arrows) in Rag1⁻/⁻E_VH1⁻/⁻ pro-B cells. **b** Diagram of the Igh locus with genomic distances and FISH probe indicated. Chromatin loops detected in pro-B cells (see Fig. 3a). **c–e** Source data are provided as a Source Data file. **c** Representative nuclei from Rag2⁻/⁻ and Rag2⁻/⁻ E_VH1⁻/⁻ CD19⁺ pro-B cells. Short FISH probes, Alexa Fluor 488 (Eμ, green), and Alexa Fluor 647 (E_VH1, blue), Alexa Fluor 555 (E_VH2, red) were hybridized simultaneously to fixed cells in duplicate. Scale bar, 2 μm. **d** Quantitation of 3D probe configurations in three-color DNA FISH. *N* = 400 alleles/genotype in two independent experiments. (*Upper panel*) Nine probe configurations. (*Lower panel*) The frequency of each probe configuration. *P* values from two tailed Chi square test. **e** Median spatial distance was derived for probes engaged in 3-way overlaps shown in panel **d** inset a. Rag2⁻/⁻ (*n* = 33), Rag2⁻/⁻E_VH1⁻/⁻ (*n* = 54) alleles. Box plots show median (middle line) spatial distances, 25th and 75th percentile (box) and 10th and 90th percentile (whiskers), and outliers (single points) between. *P* values from two-tailed Wilcoxon signed rank sum test. **f** Graphically displayed median spatial distances between probes in three-way overlaps from Rag2⁻/⁻ (black arrows) and Rag2⁻/⁻E_VH1⁻/⁻ (gray arrows) pro-B cells. **g** Representation of Igh alleles in distinct spatial configurations (**a–c**, **e–g**, **h**, **i**).

all nine possible FISH probe configurations (Fig. 7b, c). Indeed, the median spatial distances for pure pairwise Eμ:E$_{VH}$2 contacts were significantly increased, and both E$_{VH}$1:E$_{VH}$2 and Eμ:E$_{VH}$1 probe distances were modestly decreased in Rag2$^{-/-}$E$_{VH}$1$^{-/-}$ pro-B cells, consistent with the virtual 4C profile (Fig. 7d).

We explored the proposition that E$_{VH}$1 participates in a multi-enhancer network by examining three-way probe overlaps (≤ 0.3 μM) that identify Eμ·E$_{VH}$1·E$_{VH}$2 molecular contacts (Fig. 7d, inset a-d). Two types of linear arrays (Eμ·E$_{VH}$1·E$_{VH}$2; Eμ·E$_{VH}$2·E$_{VH}$1) were significantly diminished in Rag2$^{-/-}$E$_{VH}$1$^{-/-}$ pro-B cells indicating that these configurations are dependent upon E$_{VH}$1 (Fig. 7d, insets b, c). The E$_{VH}$2·Eμ·E$_{VH}$1 configuration occurs at very low frequencies and was not further considered (Fig. 7d, insets d). Although the frequency of superimposed Eμ·E$_{VH}$1·E$_{VH}$2 probe overlaps increased, the internal median spatial distances between probes were perturbed in Rag2$^{-/-}$E$_{VH}$1$^{-/-}$ pro-B cells suggesting a deformed topology (Fig. 7d, inset a, 7e–g). These studies confirm that in the absence of E$_{VH}$1, 1) E$_{VH}$2 becomes the preferred Eμ interaction partner and 2) the enhancer hub frequency is decreased or its internal conformation is altered.

### E$_{VH}$1 is a regional regulator of Igh gene expression

To assess E$_{VH}$1 enhancer function we evaluated transcriptional activity for a set of Igh index genes in Rag2$^{-/-}$ and Rag2$^{-/-}$E$_{VH}$1$^{-/-}$ pro-B cells. C$_H$ germline transcripts (GLTs) (IμCμ, Iγ2bCγ2b), V$_H$ GLTs (V$_H$14-2, V$_H$J558), antisense (AS) transcripts from intergenic regions (DQ52, 81X, 7183, J606, J558), and the PAIR4 lncRNA were analyzed in RT-PCR assays (Fig. 8a)[40,50,51]. As in the 445.11 E$_{VH}$1 KO lines, the V$_H$14-2 GLT was significantly reduced in Rag2$^{-/-}$ E$_{VH}$1$^{-/-}$ pro-B cells (Fig. 8b). Likewise, intergenic AS transcripts DQ52, V$_H$81X, and 7183 were significantly diminished in Rag2$^{-/-}$E$_{VH}$1$^{-/-}$ pro-B cells indicating E$_{VH}$1 dependency (Fig. 8b). In contrast, the expression of other genes was unchanged (Iγ2bCγ2b GLT, PAIR 4 lncRNA) or elevated (sense V$_H$J558, intergenic AS J606, J558) in Rag2$^{-/-}$E$_{VH}$1$^{-/-}$ pro-B cells (Fig. 8b, c). The Iγ2bCγ2b GLT and PAIR 4 lncRNA are located downstream of Eμ and upstream of E$_{VH}$3 are apparently outside the E$_{VH}$1 ZOI (Fig. 8b). Elevated expression of sense spliced and unspliced V$_H$J558 transcripts and AS intergenic J606 and J558 transcripts were detected relative to the 18 S RNA loading control in Rag2$^{-/-}$E$_{VH}$1$^{-/-}$ pro-B cells indicating dysregulated transcription (Fig. 8b, c). Reduced expression of GLTs in the E$_{VH}$1 ZOI correlates with the decreased V$_H$ to DJ$_H$ rearrangement. The upregulated transcripts are all located upstream of the E$_{VH}$1 ZOI and in a region of the locus which experiences hyper-contraction and increased proximity with Eμ providing a plausible explanation for increased transcription (Fig. 6e, f). However, because the number of genes tested for expression is relatively small the correlation of transcription regulation with the E$_{VH}$1 ZOI is an inference.

### E$_{VH}$1 restricts chromatin contacts in the ZOI

To determine whether E$_{VH}$1 anchored chromatin contacts influence V$_H$ gene usage in the ZOI, we divided the locus into 10 kb bins and mapped all 87 rearranging V$_H$ genes identified in the VDJ-seq analysis into these bins. Next, virtual 4C contacts using the E$_{VH}$1 and Eμ baits were mapped and those contacts that fall within these same 10 kb bins were identified. Comparison of rearranged V$_H$s (top panel) to virtual 4C contact profiles revealed that the cumulative frequency of E$_{VH}$1 anchored contacts was significantly elevated in the ZOI (dashed red box) and randomly perturbed outside of this region in Rag1$^{-/-}$E$_{VH}$1$^{-/-}$ pro-B cells (Fig. 8d, e). In contrast, Eμ anchored interactions displayed no discernable regional contact pattern in Rag1$^{-/-}$E$_{VH}$1$^{-/-}$ pro-B cells (Fig. 8d, e). Thus, in the absence of E$_{VH}$1, increased chromatin interactions are inversely correlated with reduced V$_H$ gene usage within the ZOI. We propose that E$_{VH}$1 underpins an architectural structure that promotes ZOI V$_H$ gene usage in part by coordinating and constraining long-range chromatin interactions.

## Discussion

We report a major role for E$_{VH}$s and TEs in configuring Igh locus topology and in V(D)J recombination. E$_{VH}$1, the V$_H$14-2 Pr, and F.6 CBE are regulators of V- > DJ recombination potential in Abl-t pro-B lines. Deletion of E$_{VH}$1 in mice led to regionally reduced V$_H$ gene usage in over a third of the V$_H$ domain, and the V$_H$ genes closer to E$_{VH}$1 were more significantly impacted. Likewise, expression of germline transcripts from several V$_H$ genes in Abl-t E$_{VH}$1 KO lines and a set of Igh index genes within the ZOI on Rag2$^{-/-}$E$_{VH}$1$^{-/-}$ pro-B cells were downregulated demonstrating a concordance of regional transcription regulation with V(D)J recombinational potential. E$_{VH}$1 is a modulator of locus conformation in both Abl-t pro-B lines and in mice. Deletion of E$_{VH}$1 led to locus decompaction of subTAD A in Abl-t pro-B lines and perturbed locus architecture in the E$_{VH}$1 ZOI in mice, identifying a link between D$_H$ proximal V-DJ recombination, Igh gene transcription, and locus conformation. Deletion of E$_{VH}$1 also led to locus-wide hyper-compaction in both Abl-t pro-B lines and in mice.

There are several possible explanations to account for the observations of locus de-contraction around E$_{VH}$1 despite hypercontraction overall. Our Hi-C data sets indicate that chromatin folds within subTAD C and involving E$_{VH}$2-E$_{VH}$4 may form independently of subTAD A-B that is structured by IGCR1- E$_{VH}$1 or IGCR1-E$_{VH}$1-E$_{VH}$2 interactions with E$_{VH}$2 as the locus nexus. This view might imply that upon E$_{VH}$1 deletion the Igh locus should become decompacted overall since E$_{VH}$1 anchored chromatin contacts with E$_{VH}$2 are lost. However, there are two factors that may mitigate this outcome. First, we note compensatory chromatin contacts that occur in E$_{VH}$1 deleted loci such as new Eμ-E$_{VH}$2 contacts in Rag1$^{-/-}$E$_{VH}$1$^{-/-}$ pro-B cells that might change locus conformation but would still contribute to overall locus compaction. Second, cohesin-mediated loop extrusion is also an important driver of chromatin loop formation[19,22,44] and enhancers can block the extrusion progress[19]. E$_{VH}$1 loss could lead to more processive loop extrusion that in turn generates greater locus compaction. A direct assessment of loop extrusion frequency in the Igh locus will be required to directly test this notion.

Our discovery of at least four E$_{VH}$s located among the intermediate and distal V$_H$s was based on their epigenetic profile. Many enhancers regulate transcription with their cognate Prs via spatial proximity created by long-range interactions[52,53], a feature displayed by E$_{VH}$1. However, Igh genes may be regulated by more than one enhancer as intergenic antisense J558 transcript expression is Eμ independent[54] and we propose that this role may be filled by one or several E$_{VH}$s. Accordingly, V$_H$ expression at the D$_H$ distal end of the locus was upregulated upon E$_{VH}$1 deletion suggesting the presence of another regional enhancer such as E$_{VH}$2, 3 or 4. Detection of an E$_{VH}$ interactome centered on E$_{VH}$1 was based on interactions with Eμ in the RC and with E$_{VH}$2-4 in the more distal regions of the locus and was confirmed in single-cell imaging studies. Competition between Eμ and the E$_{VH}$s is implied by the preferential formation of Eμ-E$_{VH}$2 contacts in Rag2$^{-/-}$E$_{VH}$1$^{-/-}$ pro-B cells. Notably, elevated (>2 fold) usage of two V$_H$ genes within the E$_{VH}$1 ZOI in E$_{VH}$1$^{-/-}$ pro-B cells implies that rearrangement of some V$_H$ genes within the same loop domain is differentially regulated by as yet undefined mechanisms.

Our studies provide evidence that architectural "stripes", formed by cohesin-mediated loop extrusion[44], initiate from both the 3' and 5' ends of the Igh locus in pro-B cells. Stripes form when one subunit of cohesin stalls near a strong CTCF loop anchor and the second subunit progressively extrudes the chromatin loop to form a multiplicity of contacts and that ultimately brings the far end of the locus into spatial proximity with the stall site and forms a loop domain[44]. These loop domains could provide a platform to "reel-in" V$_H$s toward the RC from various positions across the locus. E$_{VH}$1 is a stripe origin that extends in the 3' direction and ultimately intersects with IGCR1, creating a loop domain encompassing the proximal and intermediate V$_H$s and promoting spatial proximity to the RC. A RAG scanning event initiating

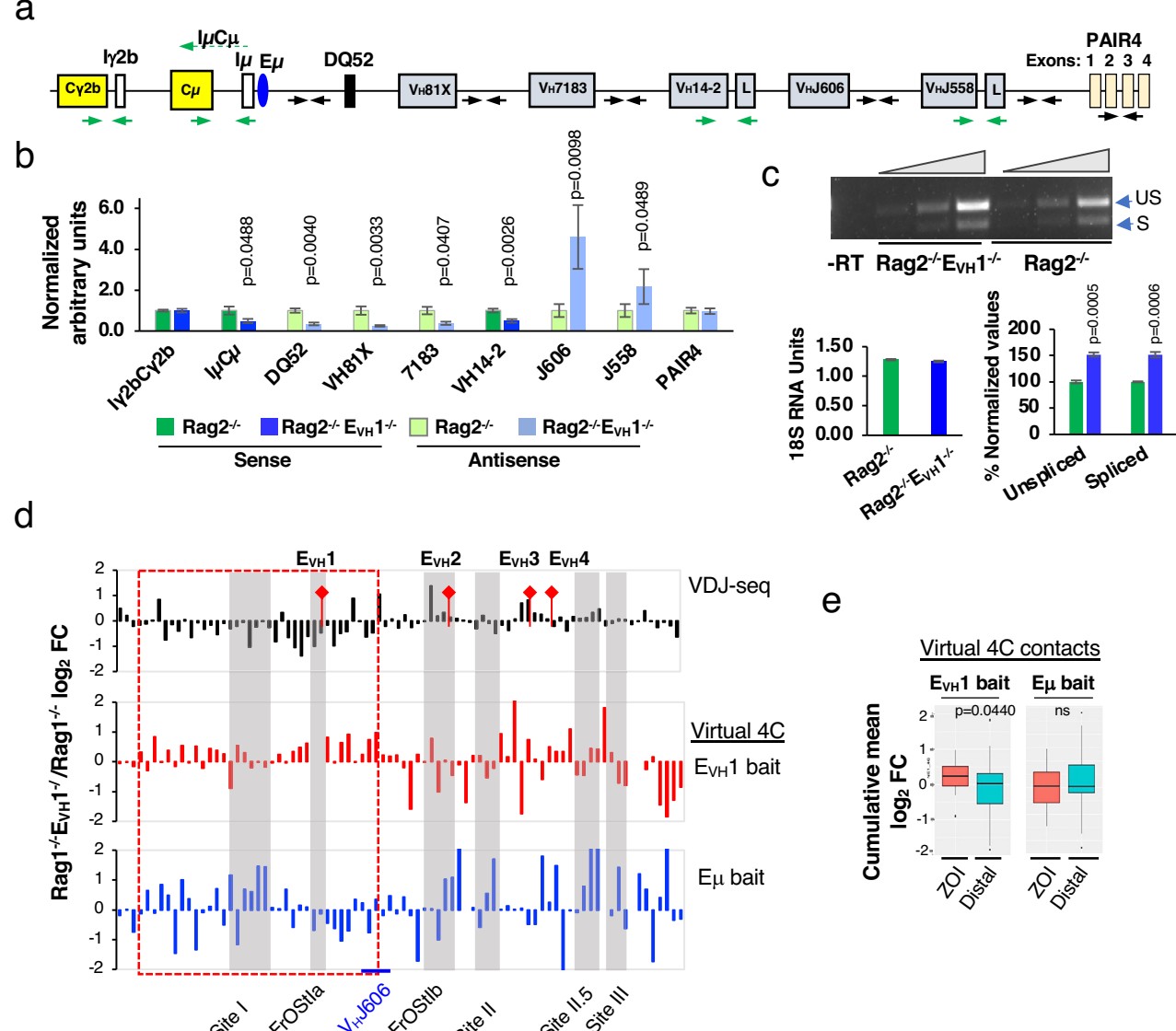

**Fig. 8 | E_VH1 regulates Igh gene expression. a** Igh genes with primers for sense (green arrows) and intergenic antisense (black arrows) transcripts. **b, c** Source data are provided as a Source Data file. P values from unpaired two tailed Student's *t* test. **b** QRT-PCR analyses from Rag2$^{-/-}$- and Rag2$^{-/-}$E_VH1$^{-/-}$- pro-B cells. RNA samples were isolated from each mouse and two cDNAs were synthesized from each RNA, tested in duplicate or triplicate and averaged. +/-SEMs are shown. RNA samples for Rag2$^{-/-}$: primers IµCµ, PAIR4, 7183 (n = 7); 81X, VH14-2, DQ52 (n = 6); J606, J558 (n = 5); Iγ2bCγ2b (n = 4) mice. Rag2$^{-/-}$E_VH1$^{-/-}$: Primers Iγ2bCγ2b, VH14-2, DQ52, J558 (n = 6); IµCµ, PAIR4, 81X, 7183, J606 (n = 4) mice. **c** Representative semi-quantitative RT-PCR for sense V_HJ558 (unspliced (US), 465 bp; spliced (S) 382 bp) (*upper panel*) using an 18 S RNA loading control (*lower left panel*). No reverse transcription (-RT). V_HJ558 PCR products were quantitated by densitometry. The values for Rag2$^{-/-}$ samples were normalized to 100. +/-SEMs. Samples from Rag2$^{-/-}$ and Rag2$^{-/-}$E_VH1$^{-/-}$ (n = 3) mice. **d** Comparison of the distribution of normalized rearranged V_H genes (upper panel, VDJ-seq analysis; Fig. 4b) with normalized virtual 4C reads for E_VH1 (middle) and Eµ (bottom) viewpoints (Fig. 6a) shown as the Rag1$^{-/-}$E_VH1$^{-/-}$/ Rag1$^{-/-}$ log$_2$ fold change (FC). Eighty-seven rearranged V_H genes mapped into 82 bins of 10 kb. E_VH1 ZOI (dashed red box). **e** Cumulative frequency of virtual 4C contacts in the ZOI and distal region is shown in box plots indicating the mean (middle line), 25th and 75th percentile (box) and 10th and 90th percentile (whiskers), and outliers (single points). *P* value from unpaired two-tailed Mann Whitney U test.

within the IGCR1-E_VH1 loop domain would create V_H gene proximity with the RC. Loss of the IGCR1- E_VH1 chromatin loop might preclude efficient RAG scanning through this region of the locus.

E_VH1 affects regional transcription and V_H to DJ_H rearrangement as well as overall locus conformation and loop extrusion. We favor the view that the participation of E_VH1 in structuring the Igh locus through chromatin looping and loop extrusion is a key to determining its regional influence on V-DJ rearrangements. However, it is difficult to exclusively assign any one aspect of E_VH1 function to its influence on V-DJ recombination, as some features, such as transcription may be necessary but not sufficient.

Recent studies identified the E88 enhancer as a structural organizer of the Igk locus, and, like E_VH1, its deletion negatively impacts rearrangement of nearby V_k genes[55]. However, E88 is distinct from E_VH1 as it affected transcription only within the region in which V_k gene rearrangement is impacted[55], whereas E_VH1 influences GLT expression throughout the V_H locus and is a component of an enhancer hub. This difference may also be related to the structural roles E_VH1 plays within the Igh locus as an architectural stripe origin and a modulator of locus compaction.

Deletion of E_VH1 results in a reduction of V_H11-2 rearrangement, the V gene most predominantly observed in B1a cells from C57Bl/6

mice[41]. These $V_H$-D-J rearrangements, with similar CDR3 sequences, are greatly expanded in B1 cells presumably by their cognate antigen PtC, and accordingly, the $E_{VH}1$ deletion results in a decrease in splenic B1a cells which bind PtC. Why do we therefore not see a decrease in B1a B cells from the PerC? One explanation could be that $E_{VH}1$ does not have as much influence in fetal vs adult life, and most B1 cells in the PerC arise from fetal precursors[56]. Alternatively, strong pressure for expansion of PtC-binding B1 cells in the PerC might lead to expansion of the residual $V_H11$-2 precursors with the relevant CDR3 sequence. In contrast, the splenic B1a repertoire is more diverse and thus presumably has less selection pressure for $V_H11.2^+$ B1a cells. Regardless, these data show a biological effect of $E_{VH}1$ deletion on a protective common antibody response and that perturbation of the preselected IgH repertoire can be reflected in the periphery.

## Methods

### Mice, cell lines and cell culture

C57BL/6 (WT) or Rag deficient mice on the C57BL/6 background were purchased from Jackson Laboratories and $E_{VH}1^{-/-}$ mice were constructed using genomic editing and then maintained in colonies at the University of Illinois College of Medicine, or Scripps Research. All procedures involving mice were approved by the Institutional Animal Care Committee of the University of Illinois College of Medicine, and the Scripps Research Institute, in accordance with protocols approved by the UIC and Scripps Research Institute Institutional Animal Care and Use Committees. Mice were housed in sterile static microisolator cages on autoclaved corncob bedding with water bottles. Food was irradiated (Envigo 7912), water was autoclaved and both were provided *ad libitum*. The standard photoperiod is 14 hours of light and 10 hours of darkness for mouse rooms. Mice receive autoclaved nesting material to enrich their environments. Cage bedding is changed in either a biosafety cabinet or a HEPA-filtered animal transfer station at least weekly. Housing density and cage size are consistent with the recommendations of the *Guide for the Care and Use of Laboratory Animals*. The ambient temperature and humidity of the rodent housing rooms are consistent with the recommendations of the *Guide for the Care and Use of Laboratory Animals*.

Bone marrow (BM) was collected from humerus, tibia and femur bones of WT and $E_{VH}1^{-/-}$ mice. Rag deficient CD19 + pro-B cells were isolated from BM using anti-CD19 coupled magnetic beads (Miltenyi) and cultured in the presence of IL7 (1% vol/vol supernatant of a J558L cell line stably expressing IL7) for 4 days. The Abelson-MuLV transformed (Abl-t) pro-B cell line, 445.3 (Rag1−/−) on the C57Bl/6 background was kindly provided by Dr. B. Sleckman (University of Alabama at Birmingham)[57]. A newly derived subclone, 445.3.11 from the Abl-t 445.3 line were cultured in RPMI 1640 (Cellgro), 10% (v/v) FBS, 4 mM glutamine (Gibco), 1 mM sodium pyruvate (Gibco), 1X nonessential amino acid (Gibco), 5000 units/ml Penicillin and 5000 mg/ml Streptomycin (Gibco), 50 mM β-mercaptoethanol (Sigma) and maintained at approximately 5x10e5 cells/ml. Splenic T cells were enriched using Mouse T Cell Enrichment Columns (MTCC-5; R&D Systems) and cultured at a density of $5 \times 10^5$ to $1 \times 10^6$, stimulated in RPMI 1640 and glutamine (4 mM) and penicillin-streptomycin supplemented with FCS (10% v/v), and activated with Con A (5 ng/ml; 15324505; MP Biomedicals).

### Flow cytometry and FACS

FACS isolation of bone marrow derived pro-B cells: Total bone marrow (BM) was collected from humerus, tibia and femur bones from groups of at least 3–10 mice. CD19$^+$ cells were isolated from BM using CD19-conjugated MACS beads and were preincubated with Fc Block for 5 minutes and then stained with specific antibodies. Isolation of pro-B cells for VDJ-seq: Mice: WT ($n$ = 3), $E_{VH}1^{-/-}$ ($n$ = 2). CD19$^+$ cells were stained with stained with Abs to B220-APC-Cy7, CD19-FITC, CD93-PE-Cy7, CD2-PE, CD43-BV421 and IgM

Fab-AF647 and were sorted on a BD FACSAria II using BD FACS Diva 6.1.3 software. Sorted pro-B cells (CD19$^+$CD93$^+$IgM$^-$CD2$^-$CD43$^+$) were used to isolate gDNA. Antibodies were used at $1 \mu l/1 \times 10^6$ cells in 1 ml up to $30 \times 10^6$ cells except for IgM ($0.5 \mu l/1 \times 10^6$ cells). Isolation of pro-B cells for qPCR and repertoire analysis: CD19$^+$ cells were stained with antibodies to CD19-PerCP (dil 1/40), B220-APC-Cy7 (dil 3/100), CD93-PE-Cy7 (AA4.1; dil 1/50), CD2-PE (dil 1/50), CD43-APC (S7; dil 1/100) and IgM-e450 (dil 1/50) and were sorted on a BD FACSAria Fusion 5-18 using BD FACS Diva 8.0.1 software. FACS purified pro-B cells (CD19$^+$B220$^+$CD93$^+$ IgM$^-$CD2$^-$CD43$^+$) were used to isolate gDNA for qPCR and repertoire analysis shown in Fig. 4f, h. Flow cytometry analyses were performed using live pro-B cells ($5 \times 10^{-5}$) washed in PBS plus 2% FCS, and stained with antibodies (CD19-PerCp (dil 1/40), B220-APC Cy7 (dil 3/100), CD93 PE-Cy7 (dil 1/50), CD43-APC (dil 1/100), CD2-PE (dil 1/2500), IgM-e450 (dil 1/100)) by gating for Fixable Viability Stain 510 (FVS510; dil 1/5000) (BD Biosciences) and/or forward- and side scatter on an Attune NxT Acoustic Focusing Cytometer lasers B, R, V, Y using NxT software version 2.7.0 (ThermoFisher). Assessment of splenic marginal zone (MZ) and follicular (Fo) B cells from mice was performed by flow cytometry. Splenocytes were isolated and the live cells were enumerated by using the viability stain FVS-510 (dil 1/5000), and the antibodies B220-APC Cy7 (dil 3/100), CD21-FITC (dil 1/100), CD23-biotin (dil 1/50) and streptavidin-APC (dil 1/500). Live splenic B1 B cells were analyzed by treating with viability stain FVS-510 (dil 1/5000), and with antibodies to CD19-PerCp (dil 1/40), B220-APC Cy7 (dil 3/100), CD23-FITC (dil 1/50), anti-IgM-e450 (dil 1/100), IgD AF700 (dil 1/50) or with CD19-PerCp (dil 1/40), B220-APC Cy7 (dil 3/100), CD43-APC (dil 1/100), IgM-e450 (dil 1/100), IgD AF700 (dil 1/50) and PtC liposomes (FITC) (dil 1/2000) and analyzed on a Attune NxT Acoustic Focusing Cytometer with gating as described[58,59] with FlowJo 10.8.1 or Summit 4.3 software. All antibodies and reagents are listed in (Supplementary Table 1).

### Quantitative RT-PCR and ChIP

RNA extraction was performed from $2–3 \times 10^6$ cells using TRIzol (Life Technologies) according to manufacturer instructions. RNA samples were further treated with the DNase I kit (Invitrogen) to remove contaminating DNA per the manufacturer's instruction. First-strand cDNA synthesis was done with RNA (1–4 μg) and the SuperScript III Reverse Transcriptase kit (Invitrogen) according to the manufacturer's instructions. Quantitative (q) real-time RT-PCR was performed using primers (Supplementary Table 2)[40] and Fast SYBR Green PCR Mix (Applied Biosystems) and Real-Time PCR System viiA7 (Applied Biosystems) as described[60] except that primers for 18 S rRNA were used[61] to normalize samples. Semi-quantitative RT-PCR assays for $V_HJ558$ transcripts were carried out using Platinum Taq DNA polymerase (Invitrogen) (25 μl) at 94 °C, 30 s for 1x; 94 °C 30 s, 60 °C 30 s, 72 °C 30 s, for 29x, 31x and 33x and 7 μl were analyzed by gel electrophoresis. The 18 S loading control was assessed by qRT-PCR. Semi-quantitative RT-PCR assays were carried out for the $V_H81X$ and $V_H2$-5 genes using the same conditions as for $V_HJ558$ except that the PCR products were harvested at 32 cycles and for 18 S rRNA at 28 cycles. All results represent the average of at least three independent experiments from 3–5 mice for each genotype or 2 biallelic identical 445.3.11 KO subclones. Each sample was assayed in triplicate and SEMs were calculated. ChIP assays for CTCF binding were performed with anti-CTCF antisera as described previously (Supplementary Table 1)[62].

### CRISPR-Cas9 mediated genomic editing of cell lines and mice

The optimal Guide (g)RNA sequence closest to the genomic targets were identified using a combination of the http://crispr.mit.edu design tool and BLAST. GRNAs were cloned into the pX330 vector at the BbsI site (Supplementary Table 3)[63]. GRNA efficiency was tested in

HEK293T cells using previously described protocol[64]. The Abl-t 445.3.11 cell line was co-transfected with gRNAs plasmid constructs (1 μg/each) together with a pmax GFP plasmid (1 μg) (Lonza) using the Amaxa Cell Line Nucleofector Kit (Lonza) (Y001 program) and a Nucleofector II (model AAD-1001N) according to the manufacturer's instructions. Cells were allowed to recover for 48 h and then GFP+ cells were purified by FACS (MoFlo Astrios) supported by Summit software (version 4.3) (Beckman Coulter, Indianapolis, IN). Cells were submitted to limiting dilution, subclones were expanded for 10–14 days and gDNA was harvested using the DNA spooling method. Subclones were screened for insertion/deletions (indels) by PCR using Platinum Taq DNA polymerase (Invitrogen) and target specific primers at 94 °C, 30 s for 1 cycle; 94 °C 30 s, 58–66 °C 30 s, 72 °C 30–60 s for 35 cycles; 72 °C 5 min for 1 cycle (Supplementary Table 4). PCR products were examined by Sanger sequencing to authenticate CRISPR/Cas9 indels.

To construct $E_{VH}1^{-/-}$ mice, Cas9 mRNA and sgRNAs $E_{VH}1$ g1 and $E_{VH}1$ g2 were injected into pronuclei of mouse zygotes at the Scripps Research Institute Mouse Genetics Core facility (Supplementary Table 3)[64]. The $E_{VH}1^{-/-}$ mouse contains a 515 bp deletion (chr12: 114182511-114183025, mm10) spanning $E_{VH}1$.

### V(D)J recombination assays in Abl-t pro-B lines

Abl-t 445.3.11 $Rag1^{-/-}$ cells were transfected with pMSCV-IRES-Bsr-Rag1 plasmid (a gift from Dr. D. Schatz, Yale University) using an Amaxa Cell Line Nucleofector Kit (Lonza) (Y001 program) and a Nucleofector II (Lonza, model AAD-1001N) according to the manufacturer's instructions. Stable transfectants were selected in 20 μg/ml blasticidin for 10–14 days and subsequently maintained in 10 μg/ml blasticidin. Cells ($1 \times 10^7$) were treated with STI-571 (2.5 μM) for 48 hours and gDNA and RNA were isolated. RNA was prepared from cells ($9 \times 10^6$) using TRIzol (Life Technologies) following the manufacturer's instructions and gDNA was prepared from cells ($1 \times 10^6$) by spooling. $D_H$->$J_H$ rearrangement was assayed using gDNA (50 ng) with DFL16.1 F and JH1R primers and the Fast SYBR green master mix (Applied Biosystems) with qPCR of the Mb1 gene as the loading control. V->DJ recombination was assayed using cDNA (2 μl) in Fast SYBR Green PCR Mix (10 μl reaction volume) in a viiA7 system (Applied Biosystems) with 18 S RNA as a loading control. Primers are listed in Supplementary Tables 2 and 5.

### DNA FISH and nuclear volume analyses

DNA probes for fluorescent in situ hybridization (FISH) were prepared from locus-specific BACs or short probes. All genomic coordinates are chr12 (mm10). The BACs probes were RI (113813346-113989276) (BAC 373N4), RII (114705813-114886430) (BAC 70F21), RIII (115539177-115773011) (BAC 368C22 or 230L2)[17] and H14 (112907511 – 113142759) located ~44 kb 3′ of HS7 of the 3′RR (BAC RP23-201H14)[65]. Short probes (4.2–7.5 kb) were 3′Eα (113221845-113226083), Eμ (113425893–113430719), Site I.3 (113987323–113994353), $E_{VH}1$ (114184075–114191532) and $E_{VH}2$ (114600831–114606206) were generated by PCR using gDNA with primers listed in (Supplementary Table 6). Probes were labeled by nick translation in the presence of Alexa Fluor 555 (red) and 488 (green) and Alexa Fluor 647 (blue). Labeled probes were hybridized with fixed cells and FISH was performed using a Nikon W1 dual CAM spinning disk confocal microscope (Center for Advanced Microscopy and Nikon Imaging Center, Northwestern University). Serial optical sections ($n = 30$–40; each slice is 0.1 mm thick) spaced by 0.2 mm were acquired. The data sets were deconvoluted using Imaris (version 9.0) software and optical sections were merged to produce 3D images. Spatial distances between probes were measured as previously described[17,27]. Purified pro-B cells from at least two mice or two slides from cell lines of different genotypes were investigated. P values of statistical significance were calculated using Mann–Whitney U-test, Wilcoxon rank sum test and Chi square test (Supplementary Table 7).

Nuclear volumes of the IL7 expanded $Rag2^{-/-}$ pro B cells were analyzed by staining with Cell Tracker Red CMTPX fluorescent dye

(Life Technologies, C34552) and DAPI and then fixed with paraformaldehyde (4%) as previously described[22]. Cells were imaged using a Nikon W1 dual CAM spinning disk confocal microscope (Center for Advanced Microscopy and Nikon Imaging Center, Northwestern University). Serial optical sections ($n = 30$–40) spaced by 0.2 μm were acquired. Nuclear volumes were determined using Imaris 9.2.1 software for a total of 120 nuclei and represented as mean ± SEM (210.4 μm$^3$ ± 5.5).

### V(D)J-seq

Total BM cells were harvested from long bones of 4–5 mice that were 6 to 10 weeks old. CD19 + cells were isolated using anti-CD19 conjugated MACS beads (Miltenyi, Auburn CA) and then pre-incubated with CD16/32 Fc Block for 5 min and stained with antibodies for CD19-BB515, CD93-PE/Cy7, IgM Fab-AF647, CD2-PE, CD43-BV421 (Supplementary Table 1). Genomic DNA (gDNA) was isolated from purified pro-B cells (DNeasy, Qiagen) and the V(D)J-seq protocol performed while omitting the negative depletion step and using custom $J_H$ primers (Supplementary Table 8)[39,66]. GDNA was sonicated to a range of 500 bp to 1000 bp using a Covaris S2 (Covaris). Library barcoding was carried out using NEBNext Multiplex Oligos for Illumina (E7600S). Samples were paired-end 2 × 300 sequenced on an Illumina MiSeq System (San Diego, CA) at the NGS Core, Scripps Research.

Paired end sequencing data was aligned to assembled V, D, and J exons using MiXCR[67], with parameters: allowChimeras false, minSumScore 120.0, maxHists 5, relativeMinVFR3CDR3Score 0.7. V/D/J segment reference libraries were from MiXCR built in Mus Musculus library. Next, unique CDR3s were assembled using MiXCR assemble, with the parameters: minimalClonalSequenceLength 12, badQualityThreshold 20, maxBadPointsPercent 0.7, qualityAggregationType Max. The frequency of $V_H$ gene usage was then counted for 1) single $V_H$ genes, and for 2) $V_H$ gene families using a customized python script. $V_H$ gene usage data determined using MiXCR software have been deposited in the GEO database (accession number: GSE203484) (Supplementary Table 10).

### Statistics

P values were calculated by using two-tailed Student's t test, two-tailed Chi square test, two-tailed Mann-Whitney U test or two tailed Mann-Whitney Wilcoxon test as indicated. Box plots represent the distribution of spatial distances with medians or means as indicated (middle line), 25th and 75th percentile (box), 10th and 90th percentile (whiskers), and outliers (single points). Box and whisker plots were generated using GraphPad or RStudio ggplot2.

### 3C library construction and analysis

3C chromatin was prepared from CD19+ IL7 expanded $Rag2^{-/-}$ pro-B cells, the Abl-t 445.3.11 line and ConA activated splenic T cells. Optimized 3C library construction and assays for the Igh locus using Hind III were performed as described[62,68]. Briefly, $Rag2^{-/-}$ pro-B cell, the Abelson transformed (Abl-t) pro-B cell line 445.3.11 and ConA activated T cells were crosslinked using 1% formaldehyde and template concentration using the Mb1 primers was determined[60]. Quantitative PCR (qPCR) in combination with 5′FAM and 3′BHQ1 modified probes (IDT) was used to detect of 3C products and primers were designed using Primer Express software (ABI) (Supplementary Table 9). Primer and probe optimization were carried out according to the manufacturer's recommendations, (http://www3.appliedbiosystems.com/cms/groups/mcb_support/documents/generaldocuments/cms_042996.pdf). A template in which all possible 3C ligation products are present in equimolar concentration was used to control for differences in amplification efficiency between primer sets (Supplementary Table 9). The data were normalized using the interaction frequency between two fragments within the non-expressed Chr5 gene desert to facilitate sample-to-sample comparisons

(Supplementary Table 9). The relative crosslinking frequency between two *Igh* restriction fragments was calculated: $X_{Igh} = [S_{Igh}/S_{GD}]$ Cell Type/$[S_{Igh}/S_{GD}]$ Control mix. $S_{Igh}$ is the signal obtained using primer pairs for two different *Igh* restriction fragments and $S_{GD}$ is the signal obtained with primer pairs for the *GD* locus fragments. The crosslinking frequency for the *GD* fragments was set to 1 to allow sample-to-sample comparisons. Data are represented as mean ± SEM. A complete laboratory protocol for 3C is available upon request. For all qPCR 3C reactions, 100 ng of chromatin was used.

## Hi-C library construction and analyses

Genome-wide in situ Hi-C libraries were constructed from Rag1$^{-/-}$, Rag1$^{-/-}$E$_{VH}$1$^{-/-}$ pro-B cells expanded in IL7 for 4–5 days using Arima Hi-C kits (Arima Genomics, San Diego, CA) as recommended by the manufacturer. In situ Hi-C was performed using two biological replicates that yielded a minimum of 1.3 billion read pairs and 0.72 billion from pro-B cells of each genotype (Supplementary Table 11) (GEO Accession No. GSE201357). Published in situ Hi-C data sets for mouse embryonic fibroblasts (MEFs) were constructed with Arima Hi-C kits (GEO Accession No. GSE113339)[43] and data was handled in parallel with the pro-B cell data. Hi-C data have been deposited in the GEO database (accession number GSE201357).

In situ Hi-C data was processed using the Juicer pipeline (v.1.5), CPU version[69]. The pipeline was set up with BWA (0.7.15–r1140)[70] to map each read end separately to the mm10 reference genome (GRCm38). Duplicate and near-duplicate reads, as well as reads that map to the same fragment were removed. Among the remaining reads, those with mapping quality score (MAPQ) < 30 are retained. Arima-HiC-specific restriction sites (mm10_GATC_GANTC.txt) were obtained from ftp://ftp-arimagenomics.sdsc. edu/pub/JUICER CUTSITE FILES. For individual replicates, raw reads from each replicate were mapped separately to the reference genome, and then filtered by the MAPQ score. For merged replicates of each sample type, valid read pairs from both replicates were merged, mapped to the reference genome, and then filtered by the MAPQ score. Hi-C contact matrices were extracted from.hic files using the Dump command provided by a Java-based program in Juicer tools[69]. Hi-C maps were then normalized with the Knight and Ruiz (KR) matrix balancing method[71]. Heatmap resolution of all individual replicates was computed using the Juicer script (calculate_map_resolution.sh) and found to be >10 kb. The reproducibility of Hi-C data was computed using the Stratum-adjusted correlation coefficient (SCC) on Chr12 using HiCRep at different resolutions[72]. The replicates were merged and displayed at 10Kb resolution.

Extraction of virtual 4C interaction matrices: The hic files were used to generate virtual 4C viewpoints from dumped matrices generated in Juicebox. KR normalized observed read matrices were extracted at 10 kb resolution. The biological replicates had stratum adjusted correlation coefficient (SCC)[72] greater than 0.9 and were merged. The interaction profile of virtual 4C were plotted by running a rolling window of 30 kb with a 10 kb slide.

Generation of Hi-C difference maps: Experimentally measured Hi-C contact matrices of individual replicate and merged samples were quantile normalized against the Hi-C contact matrices of the uniformly sampled random ensemble of the corresponding cell type. Normalizing each sample against the target distribution of uniformly sampled random ensemble will remove the between-replicate biases in frequencies due to differences in sequencing depth[73]. Hi-C difference maps were assembled by quantile normalization of Hi-C contact frequencies using a null distribution generated using previously described methods[74,75]. The source code for the null model chromatin folding by fractal Monte Carlo is available via git repository at https://bitbucket.org/aperezrathke/chr-folder and step by step procedure to generate the null distribution is explained https://bitbucket.org/aperezrathke/chr-folder under "Null distribution" subsection. To generate the target distribution of the corresponding

cell type, we followed the approach of reference[74]. Briefly, we generated a random ensemble of 300,000 polymer chains that are sampled uniformly from all geometrically feasible polymer chains confined in the nuclear volume. Each polymer chain consists of 2000 spherical monomer beads. Individual monomer beads have a diameter of ~51 nm and represent ~10 kb of DNA, which is the resolution of the Hi-C data. To model the effects of confinement and volume exclusion, we constrained each self-avoiding polymer chain to reside within a spherical nuclear volume that is appropriate for the cell type. For pro-B cells, the nuclear volume is taken as 213 $\mu m^3$; this was scaled to ~779 $nm^3$ to accommodate a chromatin segment of 20 MB, which preserves a constant base pair density relative to the entire genome. We generated a random ensemble of 300,000 polymer chains of length 20 MB of 2000 monomers, all residing in the scaled volume. For the Igh locus, we considered the region of chr12:113,215,000–chr12:116,045,000 of 3.31 MB length. We took each chromatin polymer chain of the 20 Mb segment and randomly selected 331 consecutive monomeric units to match the size of the Igh locus. The resulting random ensemble for the Igh locus has 300,000 polymer chains, each consisting of 331 spherical monomer beads. For MEF, the nuclear volume is taken as 881 $\mu m^{3[76]}$ this was scaled to ~534 $nm^3$ for the Igh locus of 3.31 MB. We then directly generated a random ensemble of 300,000 polymer chains of length 3.31 MB of 331 monomers. The contact frequency matrix of randomly sampled chromatin chains was obtained by counting the frequency of interacting monomer pairs. A pair of chromatin monomers are considered as interacting, if their Euclidean distance is ≤80 nm[77]. Contact matrices of each pro-B cell KO sample were quantile normalized against the contact matrices of the uniformly sampled random ensemble of the corresponding cell type. These quantile normalized frequencies of each sample are then compared with each other, from which differential plots are obtained to identify differences between the two biological samples. Specifically, quantile normalized frequencies of MEF are subtracted from quantile normalized frequencies of each pro-B cell KO sample.

## Reporting summary

Further information on research design is available in the Nature Portfolio Reporting Summary linked to this article.

## Data availability

The data that support this study are available from the corresponding author upon reasonable request. The Hi-C and VDJ-seq data sets generated in this study are available in the NCBI Gene Expression Omnibus (GEO) repository under accession numbers GSE203484, GSE201357. Public ChIP data sets used in this study were: GSM1635413, GSM1635411, GSM546523, GSM1038263, GSM2255552, GSM546527, GSM539537, GSM987808, GSM4350110, GSM4805384, GSM1296534, GSM1156665, GSM1156667, GSM1006370 and the mouse reference genomes: mm10 (GRCm38) and mm9 (GSCv37). DNA FISH distance measurements, qPCR, and qRT-PCR are provided in the Supplementary/Source Data file. All raw FACS data is available upon request to the corresponding author and will be provided within four weeks from the request data. Source data are provided with this paper.

## Code availability

Source code for the null model chromatin folding by fractal Monte Carlo is available via the git repository at https://bitbucket.org/aperezrathke/chr-folder and step by step procedure to generate the null distribution is explained https://bitbucket.org/aperezrathke/chr-folder under "Null distribution" subsection.

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

## Acknowledgements

This work was supported by grants from the NIH to A.L.K. and A.J.F. (RO1AI121286), to A.L.K. (R21AI133050) and to J.L. (R35 GM127084). We thank Drs. H. Shen, S. Bhat, F. Grigera and Ms. E. Drake for expert technical assistance. The authors declare that they have no competing financial interests.

## Author contributions

A.L.K. and A.J.F. conceptualized the research. K.H.B., S.P., X.Q., E.K., J.X., J.F.C., and R.W. designed and performed experiments. S.N., H.F., X.L., and J.L. performed data analyses and computation., N.B. provided resources. A.L.K., K.H.B., and S.P. wrote the manuscript. A.L.K., A.J.F., S.P., and S.N. edited the manuscript., A.L.K. and A.J.F. supervised the study, and all authors contributed to and approved the manuscript.

## Competing interests

The authors declare no competing interests.
