## [Peer Review File · Nature Communications]

REVIEWER COMMENTS

Reviewer #1 (Remarks to the Author):

This paper examines the important question of how large arrays of gene segments spanning Mb distances can be activated for V(D)J recombination. The paper also addresses the interrelationship of chromatin loop extrusion/RAG scanning and architectural elements that establish loop boundaries in regulating recombination. In particular, how do these two mechanisms interact to determine Vh gene segment usage? The authors identify four DNA elements in the Igh V region based on epigenetic and transcription factor binding parameters that they refer to as novel enhancers (NE), and the paper focuses largely on NE1 and Vh14-2, a Vh gene segment marked by highly active chromatin and transcription. They demonstrate that these elements are involved in numerous looping interactions and then study the effects of deletion of these elements in cell lines and mice on chromatin architecture, transcription and recombination of Igh. These experiments demonstrate that these elements are involved in an interdependent set of looping interactions, some CTCF dependent and others not. This hub of interactions appears to regulate germline transcription of multiple V gene segments over distances of hundreds of kb. A particularly interesting finding is that deletion of one element (e.g., the Vh14-2 promoter) can disturb interactions between other elements, leading the authors to propose the existence of interdependent interaction "hubs" that help control locus function. Interactions are convincingly analyzed by both 3C (a bulk assay) and microscopy (a single cell assay). The second half of the paper focuses on analysis of the NE1 knockout mouse made by the authors. The authors characterize recombination, transcription, and chromatin looping in detail in B cells from the mice (including Hi-C analysis). While B cell development is relatively normal in NE1^{-/-} mice, they display reduced V segment germline transcription and recombination in a large domain (the NE1 zone of influence, ZOI) and increases in some other parts of the locus. Looping interactions are perturbed in manner consistent with the existence of interaction hubs. The authors identify Hi-C "stripes" running from NE1 back toward the Igh recombination center, indicative of loop extrusion anchored at NE1, an interesting observation. A particularly nice finding is that the perturbation of recombination caused by lack of NE1, while modest in magnitude (2-fold or less for most affected segments), has a biological effect in causing a reduction in a specific subset of B1a B cells that recognize phosphatidylcholine.

This data-rich paper identifies multiple novel regulatory elements, demonstrates their role in chromatin architecture, transcription, recombination, and in establishing a normal Vh repertoire and antigen binding reactivity. It is a thorough and important study that is appropriate for publication in Nature Communications. The one limitation of the paper is in mechanistic depth—in particular, it is not yet clear if/how the changes in looping/architecture caused by deletion of the various elements contributes to the changes in recombination observed. This is a difficult question that the field has been wrestling with for decades and it would not be appropriate to ask the authors to answer it here. I strongly support acceptance of the paper after the authors address a number of issues relating to the presentation of the findings, detailed below.

Specific comments:

1. I struggled with the presentation of the Hi-C data. It was difficult to see the differences identified by the authors because the labeling on the figures partially obscured them and the figures provided to the reviewers were of low resolution (problematic for Figs. 3, 5, and 6). For the region covered in Fig. 5B, the authors should show (in supplement) the Hi-C data itself for the 3 genotypes that give rise to the subtracted plots shown in Fig. 5B, so readers can see the underlying data (e.g., are the stripes identified by the authors visible in the unsubtracted data?). Even more important, the authors should show a third subtracted plot: that of RAG1-ko minus RAG1-ko-NE1^{-/-}, so that the effects of deleting NE1 can be more easily appreciated.
2. As noted above, there are limitations in the mechanistic information that can be gleaned

- from the findings. I strongly suggest a paragraph in the Discussion to address what can, and what cannot, be inferred mechanistically. In particular, is it possible that the effects on recombination caused by deletion of NE1 are explained largely or entirely by alterations in germline transcription? What, if anything, can be concluded about the significance of loop extrusion anchored at NE1? How do the findings alter our understanding of the link between architectural elements and RAG scanning—or is it too early to draw conclusions?
3. Page 8 notes “substantial decompaction” of Igh upon deletion of NE1 while on page 12, a section of Results is devoted to “locus hypercontraction” caused by loss of NE1. This appears contradictory, though I suspect that it is just that I had difficulty following the details of the findings. I strongly suggest a paragraph in the Discussion addressing contradictions (if any) in the data and summarizing what is compacted and what is decompacted upon deletion of NE1.
 4. Lines 342-345: the conclusions drawn based on Supplementary Fig. 6B, C are not convincing—the differences identified are modest at best.
 5. Lines 424-425 and Fig. 7E. The authors claim that “that NE1 contacts were significantly elevated in the ZOI”, but Fig. 7E shows that the log₂ fold change in the ZOI is centered on zero for both baits. This contradicts the claim. There are certainly no statistics provided to support a claim of significance.
 6. The sentence on lines 70-71 should be deleted or rewritten. It has already been proven that diffusion is not sufficient, so it doesn’t make sense anymore to say that diffusion alone “may be insufficient”.
 7. Fig. 1B: What is RNA-seq- and RNA-seq+? In general, the figure legends are sparse and do not define all of the symbols and abbreviations.
 8. First paragraph of Results: the authors launch into a detailed description of the data without telling the readers what type(s) of cells the ChIP-seq data were obtained from (and it’s not in the legend either) or how they identified NE1. The manuscript refers to the “discovery” of NE1, 2, 3 and 4 but doesn’t describe the discovery process. Was it an unbiased search for elements with particular characteristics? Or were these four elements noted in a visual scan of the data from the locus? Was NE1 identified in this study or had it been reported previously? The way the text is written, NE1 appears without an indication of the criteria used to select elements of interest. In general, the authors need to work harder to make the manuscript accessible to readers.
 9. Fig. 2D: please label on the figure what is being assayed.
 10. Lines 233-235 refer to “Site I.3-NE1 interactions” and Fig. 3D, but Fig. 3D doesn’t show these interactions. I think they should be referring to Fig. 3F.
 11. NE1 deletion strongly reduces Vh81X usage in the cell line assay but appears to have no significant effect in the mice; why? This discrepancy should be addressed.
 12. Why are some Vh segments in the NE1 ZOI increased in their usage in the NE1 ko?
 13. Line 396, a couple of typos.
 14. Line 164: “in the absence of ST!571 induction” should be STI-571.
 15. Line 196: “significantly diminished only in NE1 KO lines (Fig. 2I).” Fig.2I does not exist.
 16. Line 427: “pattern in Rag1-/-NE1-/- pro-B cells”, should be Rag2-/- NE-/- pro-B cells, according to Fig. 7D.

Reviewer #2 (Remarks to the Author):

Mechanisms that diversify the primary antibody gene repertoires are fundamental questions not fully understood. The 3D chromatin structure of IgH locus plays a significant role in mediating IgH V(D)J recombination. However, despite great progress on this, mechanisms of IgH locus contraction remain to be elucidated. This manuscript by Bhat et al. provides new insights into this problem from an enhancer-mediated IgH locus conformation perspective. The authors identified several IgH cis-elements, including novel

enhancers (NEs) that engage in multiplexed chromatin interaction networks across the IgH locus. They focused on one of the NEs, NE1, and demonstrated its roles in organizing the IgH locus configuration and regulating the transcription and recombination activity of VH segments using comprehensive strategies. They proposed that NE1 functions as along-range loop extrusion barrier to contribute to IgH locus contraction and further impact IgH V(D)J recombination. Of note, this study is not a duplication of Barajas-Mora et al., 2019 on the role of enhancer elements in Igk locus contraction and Vk recombination.

Overall, the work identified new cis-elements at the IgH locus and demonstrated their architectural and regulatory role in VH V(D)J recombination. As the IgH locus has been shown to be a great model for understanding high-order chromatin structure and gene activity regulation, this work should be of broad interest to a wide audience. However, I have multiple comments that, if addressed, would further improve the study:

Major points:

- 1) The authors claimed the site I.3 fragment is a major interaction partner with the NE1 and F.6 (Line 130; Fig. 1D,1E). However, from the 3C-qPCR results, locales beyond I.3 within Site I region also mediate substantial pro-B cell-specific interactions with multiple locales within FrOStla region. Thus, it is more appropriate to conclude that Site I region interacts with the FrOStla region rather than a particular locale within the respective region interacting with each other.
- 2) The authors found that deleting the VH14-2 promoter diminished the transcription of several VH segments, e.g., VH14-2, VH81X, and VH2-5 (Fig. 2D, E). The author claimed that VH14-2 Pr may contribute to NE1 function by facilitating its spatial proximity to VH exons. However, this has another plausible explanation, which is that VH14-2 Pr could be just an enhancer-like element that plays a more direct role in regulating the transcription of nearby VHs. The H3K27ac marker is the key to determining this. But this analysis is not included in Fig. 1B (comparing the left and right panels, the H3K27ac track is missing for VH14-2 Pr). The authors should clarify this.
- 3) The effects of NE1 deletion on the rearrangements of VHs, e.g., VH81X and VH7183 family members in v-Abl cells (Fig. 2H) seem much more dramatic than that in mouse pro-B cells (Fig. 4B, C). The authors should clarify this difference. Since the authors obtained high-throughput VDJ-seq data from mouse pro-B cells, they should be able to analyze the impacts of NE1 deletion on D-J and V-DJ rearrangement more carefully. They showed a PCR-based selective analysis of a few rearrangement events (Fig. 2G,2H).
- 4) The correlation between reduced VH11.2 rearrangement and diminished Ptc-binding B1a cells in NE1 ko mice is an interesting point. However, as the VH11.2 rearrangement is assayed from a pre-selection sample, the B1a cells in the spleen undergo extensive selection. To further support this point, the authors should count both the production and non-production portions of VH11.2 rearrangements in WT and NE1 ko mice using a similar analysis strategy as shown in Lin et al., 2016 and Bolland et al., 2016 papers.
- 5) The authors tried to correlate VH rearrangements, transcription, and chromatin interaction within the NE1 ZOI (Fig. 7B, 7D), which should be the critical point of the study. However, it is not that convincing to me from at least two aspects: 1) the transcription analysis is only performed on a few selected VH segments by RT-qPCR (Fig. 7B). How can this be used for correlation analysis? 2) the claimed correlation between the patterns of VH rearrangement from VDJ-seq data and chromatin interaction anchored by NE1 and Eu from virtual 4C doesn't look evident, as shown in Fig. 7D. The authors should clarify this more carefully.

Other points :

- 1) Line 147: the authors keep claiming that the deletions are "bi-allelic identical". Does this

mean the deletion is homozygous, or are the deleted sequences on both alleles IDENTICALLY the same? This is not a big deal, but it should be clarified.

2) The manuscript needs to be incredibly polished. It has many low-level issues. For example:

Line 76: "with cohesion"

Line 84: "100 CBE are"

Line 164: "ST!571"

Line 418: Where is fig. 5G, H?

Line 444: Fig. 5B, F correctly cited?

RE: NCOMMS-22-33840

Corresponding Author: A. Kenter

REVIEWER COMMENTS

We thank the Reviewers for the time they have taken with our manuscript and for their highly constructive critiques. In response to the Reviewers' critiques we further verified the VDJ-seq repertoire analyses in qRT-PCR studies for VH7183 and VHJ558 rearrangements in mice and Abl-t cell lines in the context of the NE1 KO. The clarity of the Hi-C findings were enhanced by editing the text and providing several new Hi-C related figures. The Discussion has been restructured to better consider the issue of subTAD-A decompaction versus locus-wide hyper-compaction. Finally, we have discussed our findings with respect to potential mechanistic conclusions. We have sought to answer each point fully. We believe that the revisions emerging from this process have sharpened the manuscript. All changes in the manuscript are shown by the blue fonts.

Reviewer #1

(Remarks to the Author):

This paper examines the important question of how large arrays of gene segments spanning Mb distances can be activated for V(D)J recombination. The paper also addresses the interrelationship of chromatin loop extrusion/RAG scanning and architectural elements that establish loop boundaries in regulating recombination. In particular, how do these two mechanisms interact to determine Vh gene segment usage? The authors identify four DNA elements in the Igh V region based on epigenetic and transcription factor binding parameters that they refer to as novel enhancers (NE), and the paper focuses largely on NE1 and Vh14-2, a Vh gene segment marked by highly active chromatin and transcription. They demonstrate that these elements are involved in numerous looping interactions and then study the effects of deletion of these elements in cell lines and mice on chromatin architecture, transcription and recombination of Igh. These experiments demonstrate that these elements are involved in an interdependent set of looping interactions, some CTCF dependent and others not. This hub of interactions appears to regulate germline transcription of multiple V gene segments over distances of hundreds of kb. A particularly interesting finding is that deletion of one element (e.g., the Vh14-2 promoter) can disturb interactions between other elements, leading the authors to propose the existence of interdependent interaction "hubs" that help control locus function. Interactions are convincingly analyzed by both 3C (a bulk assay) and microscopy (a single cell assay).

The second half of the paper focuses on analysis of the NE1 knockout mouse made by the authors. The authors characterize recombination, transcription, and chromatin looping in detail in B cells from the mice (including Hi-C analysis). While B cell development is relatively normal in NE1^{-/-} mice, they display reduced V segment germline transcription and recombination in a large domain (the NE1 zone of influence, ZOI) and increases in some other parts of the locus. Looping interactions are perturbed in manner consistent with the existence of interaction hubs. The authors identify Hi-C "stripes" running from NE1 back toward the Igh recombination center, indicative of loop extrusion anchored at NE1, an interesting observation. A particularly nice finding is that the perturbation of recombination caused by lack of NE1, while modest in magnitude (2-fold or less for most affected segments), has a biological effect in causing a reduction in a specific subset of B1a B cells that recognize phosphatidylcholine.

This data-rich paper identifies multiple novel regulatory elements, demonstrates their role in chromatin architecture, transcription, recombination, and in establishing a normal Vh repertoire and antigen binding reactivity. It is a thorough and important study that is appropriate for publication in Nature Communications. The one limitation of the paper is in mechanistic depth—in particular, it is not yet clear if/how the changes in looping/architecture caused by deletion of the various elements contributes to the changes in recombination observed. This is a difficult question that the field has been wrestling with for decades and it would not be appropriate to ask the authors to answer it here. I strongly support acceptance of the paper after the authors address a number of issues relating to the presentation of the findings, detailed below.

Specific comments:

1. I struggled with the presentation of the Hi-C data. It was difficult to see the differences identified by the authors because the labeling on the figures partially obscured them and the figures provided to the reviewers were of low resolution (problematic for Figs. 3, 5, and 6). For the region covered in Fig. 5B, the authors should show (in supplement) the Hi-C data itself for the 3 genotypes that give rise to the subtracted plots shown in Fig. 5B, so readers can see the underlying data (e.g., are the stripes identified by the authors visible in the unsubtracted data?). Even more important, the authors should show a third subtracted plot: that of RAG1-ko minus RAG1-ko-NE1-/-, so that the effects of deleting NE1 can be more easily appreciated.

We apologize to the Reviewers for any inconvenience. The resolution of the figures provided to the reviewers is outside of our control.

The underlying data used for the Hi-C subtraction plots was provided in Suppl. Figure 6B,C. The stripes are indeed visible in the un-subtracted data (Suppl. fig. 6B,C black arrowheads) as now noted in the text (line 318-319). To make the findings in the Hi-C difference maps easier to discern we presented an expanded version of the maps shown in Figure 6 in the new Suppl Figure 7C. We have also shown a third subtracted plot representing Rag1-/- (red intensities) subtracted from Rag1-/-NE1-/- (blue intensities) for the region spanning IGC1-NE2, as requested by the Reviewer (Suppl Figure 7D).

2. As noted above, there are limitations in the mechanistic information that can be gleaned from the findings. I strongly suggest a paragraph in the Discussion to address what can, and what cannot, be inferred mechanistically. In particular, is it possible that the effects on recombination caused by deletion of NE1 are explained largely or entirely by alterations in germline transcription? What, if anything, can be concluded about the significance of loop extrusion anchored at NE1? How do the findings alter our understanding of the link between architectural elements and RAG scanning—or is it too early to draw conclusions?

Reviewer #1 Points 2 and 3 are focused on aspects of the Discussion. We have re-structured the Discussion overall and have addressed each of the Reviewer's concerns.

We have now summarized the various functions of NE1 and considered which of these is mechanistically essential in paragraph 4 of the Discussion (line 509). We conclude,... *"We favor the view that the participation of NE1 in structuring the Igh locus through chromatin looping and loop extrusion is a key to determining its regional influence on V-DJ rearrangements. However, it is difficult to exclusively assign any one aspect of NE1 function to its influence on V-DJ recombination, as some features, such as transcription may be necessary but not sufficient."*

3. Page 8 notes “substantial decompaction” of Igh upon deletion of NE1 while on page 12, a section of Results is devoted to “locus hypercontraction” caused by loss of NE1. This appears contradictory, though I suspect that it is just that I had difficulty following the details of the findings. I strongly suggest a paragraph in the Discussion addressing contradictions (if any) in the data and summarizing what is compacted and what is de-compacted upon deletion of NE1.

To clarify: the “substantial decompaction” of the Igh locus noted on page 8 was referring to DNA FISH findings using short probes (E μ , Site I.3 and NE1) for subTAD A (0.75 Mb) in the context of Abl-t KO lines. We have now added “in subTAD A-B” to further clarify this (line 220). The section reporting “locus hypercontraction” caused by NE1 deletion was in pro-B cells using BAC probes spread across the entire locus (~2.9 Mb). We have now added additional data to show that the Igh locus also undergoes substantial compaction in the Abl-t NE1 KO line in accord to findings in primary pro-B cells (Suppl. Figure 8). These results indicate a similarity of the impact of NE1 deletion on locus conformation in mice and the Abl-t KO line. This information is added to the Results section (lines 365-369) “*Likewise, closer spatial distances are detected between the RI anchor, with RII and RIII probes and greater distance to H14 in the Abl-t NE1 KO line reflecting increased locus compaction in the distal regions of the Igh locus relative to the control (Suppl. fig. 8).*” Thus, deletion of NE1 has similar effects on compaction of the distal regions of the Igh locus in primary pro-B cells and in Abl-t KO lines.

The first two paragraphs in the Discussion have been restructured and summarize the effects of NE1 deletion on locus conformation. The second paragraph of the Discussion addresses why it is possible that decompaction in subTAD A could occur in the context of overall locus hyper-contraction as follows (line 469-481): “*There are several possible explanations to account for the observations of locus de-contraction around NE1 despite hypercontraction overall. Our Hi-C data sets indicate that chromatin folds within subTAD C and involving NE2-NE4 may form independently of subTAD A-B that is structured by IGCR1-NE1 or IGCR1-NE1-NE2 interactions with NE2 as the locus nexus. This view might imply that the upon NE1 deletion the Igh locus should become de-compacted overall since NE1 anchored chromatin contacts with NE2 are lost. However, there are two factors that may mitigate this outcome. First, we note compensatory chromatin contacts that occur in NE1 deleted loci such as new Em-NE2 contacts in Rag1^{-/-}NE1^{-/-} pro-B cells that might change locus conformation but would still contribute to overall locus compaction. Second, cohesin mediated loop extrusion is also an important driver of chromatin loop formation^{19,22,44} and enhancers can block the extrusion progress¹⁹. NE1 loss could lead to more processive loop extrusion that in turn generates greater locus compaction. A direct assessment of loop extrusion frequency in the Igh locus will be required to directly test this notion.*”

4. Lines 342-345: the conclusions drawn based on Supplementary Fig. 6B, C are not convincing—the differences identified are modest at best.

This data set has been removed from the manuscript.

5. Lines 424-425 and Fig. 7E. The authors claim that “that NE1 contacts were significantly elevated in the ZOI”, but Fig. 7E shows that the log₂ fold change in the ZOI is centered on zero for both baits. This contradicts the claim. There are certainly no statistics provided to support a claim of significance.

It is evident that we needed to clarify this section. We find that when anchored on the NE1 bait that chromatin contacts increase in the ZOI and are random outside this region in Rag1^{-/-}NE1^{-/-} pro-B cells (now fig. 8D). The cumulative frequency of virtual 4C contacts (using Mann-Whitney statistics and box and whisker plots) differ between the ZOI and outside this region (fig. 8E). In contrast, a similar analysis anchored on E μ shows random interactions across the Igh locus. We have enlarged Figure 8E to make

the differences between these segments of the locus more evident. In the box (25th to 75th percentile) and whisker (10th and 90th percentile) plots of the data points and the cross line represents the mean (p values, Mann Whitney U test).

We have edited this section (lines 437-442) and in part it now reads: *“Comparison of rearranged V_Hs (top panel) to virtual 4C contact profiles revealed that the cumulative frequency of NE1 anchored contacts was significantly elevated in the ZOI (dashed red box) and randomly perturbed outside of this region in Rag1^{-/-}NE1^{-/-} pro-B cells (fig. 8D,E). In contrast, E_μ anchored interactions displayed no discernable regional contact pattern in Rag1^{-/-}NE1^{-/-} pro-B cells (fig. 8D,E)”*. The statistics to support our interpretation are shown in Figure 8E.

6. The sentence on lines 70-71 should be deleted or rewritten. It has already been proven that diffusion is not sufficient, so it doesn't make sense anymore to say that diffusion alone “may be insufficient”.

This sentence (lines 67-70) has been rewritten as follows: *“Igh locus contraction⁶⁻⁸, diffusion related mechanisms⁹, and RAG scanning¹⁰ bring V_H segments into closer spatial proximity to the RC permitting usage of the V_H from across the locus. Nevertheless, the contribution of specific regulatory elements to locus conformation and the rearrangement of distal V_H segments remains unclear.”*

7. Fig. 1B: What is RNA-seq- and RNA-seq+? In general, the figure legends are sparse and do not define all of the symbols and abbreviations.

The RNA-seq- and RNA-seq+ refer to negative strand and positive strand analysis. This is now made clear in the figure and legend. We bolstered the figure legend descriptions overall.

8. First paragraph of Results: the authors launch into a detailed description of the data without telling the readers what type(s) of cells the ChIP-seq data were obtained from (and it's not in the legend either) or how they identified NE1. The manuscript refers to the “discovery” of NE1, 2, 3 and 4 but doesn't describe the discovery process. Was it an unbiased search for elements with particular characteristics? Or were these four elements noted in a visual scan of the data from the locus? Was NE1 identified in this study or had it been reported previously? The way the text is written, NE1 appears without an indication of the criteria used to select elements of interest. In general, the authors need to work harder to make the manuscript accessible to readers.

All NEs are newly identified in a visual scan of epigenetic marks and transcription factor binding. A point now noted on line 105. None of these elements were previously identified. VH14-2 located in Site 1 was previously noted to be highly transcribed and a reference is provided (line 106). The ChIP-seq are all from Rag deficient pro-B cells and all data sets are now listed in Suppl. Table 11.

9. Fig. 2D: please label on the figure what is being assayed.

This has now been done.

10. Lines 233-235 refer to “Site I.3-NE1 interactions” and Fig. 3D, but Fig. 3D doesn't show these interactions. I think they should be referring to Fig. 3F.

Fig. 3D has been changed to 3F.

11. NE1 deletion strongly reduces Vh81X usage in the cell line assay but appears to have no significant effect in the mice; why? This discrepancy should be addressed.

VDJ-seq allows us to discern the relative ratio of VH genes used in normalized data sets. However, this does not allow us to determine the relative level of VH gene usage between data sets as there is no external common denominator, a problem similar to comparing ChIP-seq data sets. To approach this question, we have now added qPCR data normalized to an external standard (eg the Mb1 gene) to allow sample to sample comparisons. We used a pan-specific VH7183 primers with reverse primers for each JH segment (Oudinet et al. NAR 2020) and analyzed VDJ rearrangements in both primary pro-B cells and the Abl-t control and NE1 KO lines for more direct comparisons. At the time these studies were carried out we chose to analyze VH7183 versus VHJ558 to get an overall view of what has happened locus wide in primary pro-B cells but did not analyze VH81X. VH81X is a member of the 7183 family and likely comports with the findings for VH7183. The following information has been added to the Results section (starting line 265) and see new Figure 4F-H and Suppl. Fig. 4b.

“VDJ-seq allows analysis of the relative ratio of V_H gene usage within normalized data sets but cannot determine if the overall level of rearrangement is lower in one sample compared to another. To address this question, we used pan-specific V_H7183 and V_HJ558 primers with reverse primers for each J_H segment⁴⁰ and reanalyzed VDJ rearrangements in both primary pro-B cells and the Abl-t control and NE1 KO lines (Suppl. fig. 4). V_H7183 rearrangements in conjunction with each J_H segment are significantly diminished or trended lower in NE1^{-/-} pro-B cells as compared to WT whereas D-J recombination is unaffected (fig. 4 F,G). Likewise, V_H7183 and additionally the V_H81X gene rearrangements are significantly reduced in the Abl-t NE1 KO line whereas D-J recombination is intact in accord with our findings (fig. 2G,H) (fig. 4 I,J, K). Hence, there is an overall similarity in the impact of NE1 deletion on V_H7183 gene usage in pro-B cells and Abl-t pro-B lines. In contrast, there are variable effects of NE1 deletion on V_HJ558 rearrangements with different J_H segments in pro-B cells which may reflect the variable impact of NE1 deletion on V_HJ558 usage found in the VDJ-seq study (fig. 4C,H). V_HJ558 rearrangements are very infrequent in Abl-t pro-B cell lines so cannot be measured here. Collectively, these findings indicate a consistent diminution of V_H7183 usage upon loss of NE1”.

12. Why are some Vh segments in the NE1 ZOI increased in their usage in the NE1 ko?

This is an interesting question. In response to the Reviewer’s query we have refined the Discussion as follows (line 493-495): *“Notably, elevated (>2 fold) usage of two V_H genes within the NE1 ZOI in NE1^{-/-} pro-B cells implies that some V_H genes within the same loop domain are differentially regulated by as yet undefined mechanisms.”*

13. Line 396, a couple of typos.

(now line 406) This has been corrected to, “...in Rag2^{-/-}NE1^{-/-} pro-B cells suggesting a deformed...”

14. Line 164: "in the absence of STI571 induction" should be STI-571.

(now line 165) We have made the correction.

15. Line 196: "significantly diminished only in NE1 KO lines (Fig. 2I)." Fig.2I does not exist.

This has been corrected to (fig. 2H).

16. Line 427: "pattern in Rag1^{-/-}NE1^{-/-} pro-B cells", should be Rag2^{-/-} NE^{-/-} pro-B cells, according to Fig. 7D.

This is now located in the section beginning at line 432 and is associated with Figure 8D. All the data shown in Figure 8D,E are derived from Rag1^{-/-} and Rag1^{-/-}NE1^{-/-} pro-B cells. The labeling is correct.

Reviewer #2:

Mechanisms that diversify the primary antibody gene repertoires are fundamental questions not fully understood. The 3D chromatin structure of IgH locus plays a significant role in mediating IgH V(D)J recombination. However, despite great progress on this, mechanisms of IgH locus contraction remain to be elucidated. This manuscript by Bhat et al. provides new insights into this problem from an enhancer-mediated IgH locus conformation perspective. The authors identified several IgH cis-elements, including novel enhancers (NEs) that engage in multiplexed chromatin interaction networks across the IgH locus. They focused on one of the NEs, NE1, and demonstrated its roles in organizing the IgH locus configuration and regulating the transcription and recombination activity of VH segments using comprehensive strategies. They proposed that NE1 functions as along-range loop extrusion barrier to contribute to IgH locus contraction and further impact IgH V(D)J recombination. Of note, this study is not a duplication of Barajas-Mora et al., 2019 on the role of enhancer elements in Igk locus contraction and Vk recombination.

Overall, the work identified new cis-elements at the IgH locus and demonstrated their architectural and regulatory role in VH V(D)J recombination. As the IgH locus has been shown to be a great model for understanding high-order chromatin structure and gene activity regulation, this work should be of broad interest to a wide audience. However, I have multiple comments that, if addressed, would further improve the study:

Major points:

1) The authors claimed the site I.3 fragment is a major interaction partner with the NE1 and F.6 (Line 130; Fig. 1D,1E). However, from the 3C-qPCR results, locales beyond I.3 within Site I region also mediate substantial pro-B cell-specific interactions with multiple locales within FrOStla region. Thus, it is more appropriate to conclude that Site I region interacts with the FrOStla region rather than a particular locale within the respective region interacting with each other.

We agree but want to highlight NE1 and the F.6 CBE as we focus on them in further experiments. This sentence has been amended to (line 132): "Thus, the Site I.3 fragment is a major interaction partner with FrOStla including NE1 and F.6 (CBE)."

2) The authors found that deleting the VH14-2 promoter diminished the transcription of several VH segments, e.g., VH14-2, VH81X, and VH2-5 (Fig. 2D, E). The author claimed that VH14-2 Pr may contribute to NE1 function by facilitating its spatial proximity to VH exons. However, this has another plausible explanation, which is that VH14-2 Pr could be just an enhancer-like element that plays a more direct role in regulating the transcription of nearby VHs. The H3K27ac marker is the key to determining this. But this analysis is not included in Fig. 1B (comparing the left and right panels, the H3K27ac track is missing for VH14-2 Pr). The authors should clarify this.

We agree that the VH14-2 Pr could be enhancer-like. However, this Pr is NOT marked by any appreciable H3K27Ac as is now shown explicitly in **amended Suppl. Figure 1A**. We think the most plausible explanation for the VH gene transcription profile is proximity with NE1. We added a sentence as follows: (line 107) "The presence of H3K4me3 which mark transcriptionally active Prs²⁹ and the absence of

enhancer associated H3K27Ac modifications, indicates that this element functions as a Pr (Suppl. fig. 1A)."

3) The effects of NE1 deletion on the rearrangements of VHs, e.g., VH81X and VH7183 family members in v-Abl cells (Fig. 2H) seem much more dramatic than that in mouse pro-B cells (Fig. 4B, C). The authors should clarify this difference. Since the authors obtained high-throughput VDJ-seq data from mouse pro-B cells, they should be able to analyze the impacts of NE1 deletion on D-J and V-DJ rearrangement more carefully. They showed a PCR-based selective analysis of a few rearrangement events (Fig. 2G,2H).

We have now added in qPCR data for a direct comparison of the WT and NE1^{-/-} pro-B cells and Abl-t control and NE1 KO lines (see amended Fig. 4F-K and Suppl.Fig. 4B). The following information has been added to the Results section (starting line 265). *"VDJ-seq allows analysis of the relative ratio of V_H gene usage within normalized data sets but cannot determine if the overall level of rearrangement is lower in one sample compared to another. To address this question, we used pan-specific V_H7183 and V_H558 primers with reverse primers for each J_H segment⁴⁰ and reanalyzed VDJ rearrangements in both primary pro-B cells and the Abl-t control and NE1 KO lines (Suppl. fig. 4). V_H7183 rearrangements in conjunction with each J_H segment are significantly diminished or trended lower in NE1^{-/-} pro-B cells as compared to WT whereas D-J recombination is unaffected (fig. 4 F,G). Likewise, V_H7183 and additionally the V_H81X gene rearrangements are significantly reduced in the Abl-t NE1 KO line whereas D-J recombination is intact in accord with our findings (fig. 2G,H) (fig. 4 I,J, K). Hence, there is an overall similarity in the impact of NE1 deletion on V_H7183 gene usage in pro-B cells and Abl-t pro-B lines. In contrast, there are variable effects of NE1 deletion on V_H558 rearrangements with different J_H segments which may reflect the variable impact of NE1 deletion on V_H558 usage found in the VDJ-seq study (fig. 4C,H). V_H558 rearrangements are very infrequent in Abl-t pro-B cell lines so cannot be measured here. Collectively, these findings indicate a consistent diminution of VH7183 usage upon loss of NE1."*

4) The correlation between reduced VH11.2 rearrangement and diminished Ptc-binding B1a cells in NE1 ko mice is an interesting point. However, as the VH11.2 rearrangement is assayed from a pre-selection sample, the B1a cells in the spleen undergo extensive selection. To further support this point, the authors should count both the production and non-production portions of VH11.2 rearrangements in WT and NE1 ko mice using a similar analysis strategy as shown in Lin et al., 2016 and Bolland et al., 2016 papers.

We agree that analysis of the Igh repertoire in splenic B1a cells in WT and NE1^{-/-} mice will be very interesting. However, analysis of mature B cell repertoire diversity is outside the scope of the present study which is focused entirely on the pre-selected repertoire. We respectfully decline to initiate these new experiments now to add to this manuscript.

5) The authors tried to correlate VH rearrangements, transcription, and chromatin interaction within the NE1 ZOI (Fig. 7B, 7D), which should be the critical point of the study. However, it is not that convincing to me from at least two aspects: 1) the transcription analysis is only performed on a few selected VH segments by RT-qPCR (Fig. 7B). How can this be used for correlation analysis? 2) the claimed correlation between the patterns of VH rearrangement from VDJ-seq data and chromatin interaction anchored by NE1 and Eu from virtual 4C doesn't look evident, as shown in Fig. 7D. The authors should clarify this more carefully.

Regarding point 5.1, we agree that the number of index genes tested is relatively small and the correlation of transcriptional activity with the NE1 ZOI is somewhat inferential. We have added a

sentence to reflect this (line 428): *“However, because the number of genes tested for expression is relatively small the correlation of transcription regulation with the NE1 ZOI is an inference.”*

Regarding point 5.2, the pattern of VH gene usage in the NE1 ZOI is anti-correlated with the pattern of chromatin interactions originating from the region around NE1 in the Rag1^{-/-}NE1^{-/-} context (fig. 8D). We have added a new subheading to distinguish between the conclusions arising from the transcriptional analysis and those related to the chromatin interaction profiling. We now more clearly note that a significant cumulative increase of chromatin interactions occurs in the NE1 ZOI of Rag1^{-/-}NE1^{-/-} pro-B cells (fig. 8E). We have also refined our explanation to read (line 434): *“To determine whether NE1 anchored chromatin contacts influence V_H gene usage in the ZOI, we divided the locus into 10 kb bins and mapped all 87 rearranging V_H genes identified in the VDJ-seq analysis into these bins. Next, virtual 4C contacts using the NE1 and E μ baits were mapped and those contacts that fall within these same 10 kb bins were identified. Comparison of rearranged V_Hs (top panel) to virtual 4C contact profiles revealed that the cumulative frequency of NE1 anchored contacts was significantly elevated in the ZOI (dashed red box) and randomly perturbed outside of this region in Rag1^{-/-}NE1^{-/-} pro-B cells (fig. 8D,E). In contrast, E μ anchored interactions displayed no discernable regional contact pattern in Rag1^{-/-}NE1^{-/-} pro-B cells (fig. 8D,E). Thus, increased chromatin interactions in the absence of NE1 are anti-correlated with reduced V_H gene usage within the ZOI. We propose that NE1 underpins an architectural structure that promotes ZOI V_H gene usage in part by coordinating and constraining long range chromatin interactions.”*

Other points :

1) Line 147: the authors keep claiming that the deletions are “bi-allelic identical”. Does this mean the deletion is homozygous, or are the deleted sequences on both alleles IDENTICALLY the same? This is not a big deal, but it should be clarified.

The deletion on both alleles is identical. The sentence now reads, “.. to generate identical deletions on both alleles for...”.

2) The manuscript needs to be incredibly polished. It has many low-level issues. For example:
Line 76: “with cohesion”

This has been corrected (now line 75).

Line 84: “100 CBE are”

This has been corrected to “CBEs” (now line 83)

Line 164: “STI571”

This has been corrected to “STI-571” – throughout the manuscript.

Line 418: Where is fig. 5G, H?

The reference to Figure 5 G,H was a typo – carried over from an earlier version of the manuscript. This has now been corrected to now Figure 6 E,F (now line 428)

Line 444: Fig. 5B, F correctly cited?

The reference to Figure 5B,F has been deleted.

REVIEWERS' COMMENTS

Reviewer #1 (Remarks to the Author):

The authors have done a thorough job of addressing my comments and these changes and the additional data have significantly improved the manuscript. One comment:

Line 442-443: "Thus, increased chromatin interactions in the absence of NE1 are anti correlated with reduced VH gene usage within the ZOI." This sentence is pretty confusing and I'm not even sure it says what the authors intend. It would be much clearer if rewritten as: "Thus, chromatin interactions in the absence of NE1 are anti correlated with VH gene usage within the ZOI." (the authors are asking readers to get their heads around an anti-correlation between an increase of one thing and a decrease of another; I for one don't know how to parse this, and I don't think the increase is actually anti-correlated with the decrease.)

David Schatz

Reviewer #2 (Remarks to the Author):

Thank the authors for their efforts in addressing the reviewers' comments. The revised manuscript is significantly improved now. However, a few points still fail to be addressed adequately.

- 1) The discrepancy of the impact of NE1 deletion on VH81X usage between mouse pro-B cells and vAbl cell lines is still not fully addressed. The authors provided some new qPCR data, which just solidified such discrepancy. They failed to address why the discrepancy occurred. Considering this is a crucial phenotype of NE1 deletion, this should be more clearly addressed.
- 2) Relating to point 1), the representative flow cytometry results depicted for the isolation of pro-B cells used for VDJ-seq (Suppl. Fig. 4a) and qPCR (Suppl. Fig. 4b) look very different. And these flow cytometry results both look different from what was shown in Suppl. Fig. 5, even with the same markers (CD19-PerCP v.s. B220-APC-Cy7). Are these sorting discrepancies responsible for the NE1 knockout phenotype discrepancy mentioned above? These discrepancies should be clarified.
- 3) The authors added the amended Suppl. Fig. 1A to show the absence of H3K27Ac for VH14-2 Pr and claims "The presence of H3K4me3 which mark transcriptionally active Prs 29 and the absence of enhancer associated H3K27Ac modifications, indicates that this element functions as a Pr (Suppl. fig. 1A)." Unfortunately, the amended profile of H3K27Ac to show its absence is not compelling. The level of the ChIP-seq signal for H3K27Ac on VH14-2 looks low, with the y-axis adjusted to 15. However, the signal seems still accumulated on VH14-2 compared to its neighboring regions. Can the authors unambiguously demonstrate the absence of H3K27Ac by running the MACS2 pipeline to show the signal accumulation is not a peak on VH14-2?
- 4) The direct correlation between reduced VH11.2 rearrangement and diminished Ptc-binding B1a cells in NE1 ko mice is still lacking. The authors may misunderstand my suggestion to count both the production and non-production portions of VH11.2 rearrangements in WT and NE1 ko mice. The authors don't need to perform any new experiments on mature B cells. Instead, they only need to count the VH11.2 rearrangements in the production and non-production portions from the VDJ-seq datasets from BM pro-B cells that they already have. In the current result, the reduction in VH11.2 total rearrangements doesn't necessarily demonstrate the contribution of reduced VH11.2 usage to diminished Ptc-binding B1a cells in NE1 ko mice, as the modest reduction in VH11.2 may predominately come from the non-production portions of VH11.2 rearrangement. The authors still need to make this point straight.

REVIEWERS' COMMENTS

We thank the Reviewers for their highly constructive critiques and for their enthusiasm for our work. Changes to the text are indicated by blue fonts.

Reviewer #1 (Remarks to the Author):

The authors have done a thorough job of addressing my comments and these changes and the additional data have significantly improved the manuscript. One comment:

Line 442-443: “Thus, increased chromatin interactions in the absence of NE1 are anti correlated with reduced V_H gene usage within the ZOI.” This sentence is pretty confusing and I’m not even sure it says what the authors intend. It would be much clearer if rewritten as: “Thus, chromatin interactions in the absence of NE1 are anti correlated with V_H gene usage within the ZOI.” (the authors are asking readers to get their heads around an anti-correlation between an increase of one thing and a decrease of another; I for one don’t know how to parse this, and I don’t think the increase is actually anti-correlated with the decrease.)

David Schatz

Our response:

Now line 547: We believe that changing the word “anti-correlated” to the more common parlance “inversely correlated” captures the sense of our finding. We have re-written the sentence as follows: “Thus, in the absence of NE1 increased chromatin interactions are *inversely correlated* with reduced V_H gene usage within the ZOI.”

Reviewer #2 (Remarks to the Author):

Thank the authors for their efforts in addressing the reviewers’ comments. The revised manuscript is significantly improved now. However, a few points still fail to be addressed adequately.

1) The discrepancy of the impact of NE1 deletion on V_H81X usage between mouse pro-B cells and vAbl cell lines is still not fully addressed. The authors provided some new qPCR data, which just solidified such discrepancy. They failed to address why the discrepancy occurred. Considering this is a crucial phenotype of NE1 deletion, this should be more clearly addressed.

Our response:

Thank you for pointing out that we had not sufficiently clarified this issue. We have revised this section substantially and hope that it is now clear. In our previous response to this question we noted that the results from the two techniques, VDJ-seq and qPCR, are measuring different things. In VDJ-seq we are measuring the ratio of a particular V_H gene rearrangement to rearrangement of all other V_H genes in the repertoire. This will not determine whether the absolute frequency has changed from the WT sample to the KO sample. In the qPCR we are measuring the real frequency of V_H rearrangement against an external control, thus allowing genotype to genotype comparisons. The VDJ-seq data sets allowed us to conclude that the ratio of V_H81X to all other V_H rearrangements is similar in WT and NE1^{-/-} pro-B cells. However, the absolute frequency of V_H7183 (of which V_H81X is a member) rearrangement is reduced in NE1^{-/-} compared to the WT pro-B cells (qPCR data sets). Likewise, in the Abl-t NE1 KO line V_H81X (not tested in pro-B cells) and V_H7183 were reduced albeit more severely than in NE1^{-/-} pro-B cells.

We suggest that the difference in the degree of V_H81X rearrangement frequency in the Abl-t lines and pro-B cells may be related to differences in the degree of locus contraction in these cell types. Abl-t pro-B cell lines fail to undergo locus contraction [1], mainly rearrange V_H81X and do not

rearrange distal V_H genes [2] because they fail to undergo locus contraction [1]. In contrast, locus contraction and distal V_H gene rearrangement occurs robustly in WT primary pro-B cells. Therefore, we took the reduced frequency of V_H81X rearrangement in the Abl-t NE1 KO lines as suggestive of NE1's potential influence on V_H gene usage in pro-B cells.

To clarify this rationale we revised the section on V_H rearrangement in pro-B cells when measured by qPCR (starting line 255) and newly wrote (beginning on line 267): *“We observe that V_H81X usage is not diminished in the VDJ-seq analysis of NE1^{-/-} pro-B cells in contrast to what was observed in the NE1^{-/-} Abl-t lines (Fig. 4f-k). It should be noted that the repertoire of rearrangements is very limited in Abl-t cell lines, with V_H81X being the dominant V_H gene used for rearrangement [2], presumably due to the absence of locus contraction [1] so that the repertoires are quite different in these two cell types. More relevant to this issue, VDJ-seq in pro-B cells allows analysis of the relative ratio of V_H gene usage within normalized data sets but cannot determine if the overall level of rearrangement is lower in one sample compared to another. Therefore, to address the possibility that the extent of proximal V_H gene rearrangements was lower in the NE1^{-/-} pro-B cells despite the relative usage of V_H7183 genes being similar, we used pan-specific V_H7183 and V_H558 primers with reverse primers for each J_H segment [3] and analyzed VDJ rearrangements in gDNA in both primary pro-B cells and the Abl-t control and NE1 KO lines (Supplementary Fig. 4) (Fig. 4f-k).”“We conclude that the frequency of V(D)J rearrangement is reduced for the proximal V_H genes in the NE1^{-/-} mice. Collectively, these findings indicate a consistent diminution of V_H7183 usage upon loss of NE1.”*

2) Relating to point 1), the representative flow cytometry results depicted for the isolation of pro-B cells used for VDJ-seq (Suppl. Fig. 4a) and qPCR (Suppl. Fig. 4b) look very different. And these flow cytometry results both look different from what was shown in Suppl. Fig. 5, even with the same markers (CD19-PerCP v.s. B220-APC-Cy7). Are these sorting discrepancies responsible for the NE1 knockout phenotype discrepancy mentioned above? These discrepancies should be clarified.

Our response:

Regarding the flow cytometry profiles showing the isolation of pro-B cells we wish to point out that the sorts were done in two different labs with two different FACS instruments and two different panels of fluorochromes. We hope that we have clarified that the VDJ-seq and qPCR techniques measure relative ratio of V_H gene usage within a sample versus absolute V_H gene usage relative to an external control, respectively. We also note the difference in the Igh repertoire of Abl-t lines and primary pro-B cells that is related to the state of locus contraction [1]. Therefore, we do not see a discrepancy in the impact of NE1 deletion on V_H gene usage in the Abl-t lines and primary pro-B cells.

Please also see new Suppl. figure 4b.

3) The authors added the amended Suppl. Fig. 1A to show the absence of H3K27Ac for VH14-2 Pr and claims “The presence of H3K4me3 which mark transcriptionally active Prs 29 and the absence of enhancer associated H3K27Ac modifications, indicates that this element functions as a Pr (Suppl. fig. 1A).” Unfortunately, the amended profile of H3K27Ac to show its absence is not compelling. The level of the ChIP-seq signal for H3K27Ac on VH14-2 looks low, with the y-axis adjusted to 15. However, the signal seems still accumulated on VH14-2 compared to its neighboring regions. Can the authors unambiguously demonstrate the absence of H3K27Ac by running the MACS2 pipeline to show the signal accumulation is not a peak on VH14-2?

Our response:

We cannot demonstrate the absolute absence of H3K27Ac on V_H14-2 as this modification is present albeit very low in real terms and relative to other marks (H3K4me1). A possible explanation for low levels of H3K27Ac on V_H14-2 is that this element interacts with NE1 which is decorated with H3K27Ac, as shown in 3C assays (fig. 1D, 2C). As the ChIP-seq protocol involves a formaldehyde cross-linking step, the V_H14-2-NE1 interaction could leave a residual footprint of H3K27Ac at V_H14-2. Our point is that the V_H14-2 gene is highly transcribed and its promoter has marks associated with canonical promoters with a high H3K4me3/H3K27Ac ratio that fits the epigenetic characterization of a promoter and not an enhancer. Nevertheless, we acknowledge that the distinction between promoters and enhancers can sometimes be blurred as there are “enhancer-like promoters” [4]. Therefore, we have softened the sentence in question (lines 107-109) to read, “*The presence of H3K4me3 which mark transcriptionally active Prs [5] and the **relative** absence of enhancer associated H3K27Ac modifications [6,7], indicates that this element **predominantly** functions as a Pr (Supplementary Fig. 1a).*”

4) The direct correlation between reduced VH11.2 rearrangement and diminished Ptc-binding B1a cells in NE1 ko mice is still lacking. The authors may misunderstand my suggestion to count both the production and non-production portions of VH11.2 rearrangements in WT and NE1 ko mice. The authors don't need to perform any new experiments on mature B cells. Instead, they only need to count the VH11.2 rearrangements in the production and non-production portions from the VDJ-seq datasets from BM pro-B cells that they already have. In the current result, the reduction in VH11.2 total rearrangements doesn't necessarily demonstrate the contribution of reduced VH11.2 usage to diminished Ptc-binding B1a cells in NE1 ko mice, as the modest reduction in VH11.2 may predominately come from the non-production portions of VH11.2 rearrangement. The authors still need to make this point straight.

Our response:

We thank the Reviewer for clarifying the question. We have two responses to this query. First, we understand that the reason to analyze the productive vs non-productive (P/NP) ratio is to determine if there is selection. However, we are unaware of selection at the pro-B cell stage. If we had looked at pre-B cell stage, then we could have expected some selection by pre-BCR, but we cannot think of any reason for selection for pro-B cells. Second, we cannot make the distinction requested. Our VDJ-seq sequencing results did not always extend completely through . We certainly can exclude obviously non-productive events. However, there may still be non-productive events in the putatively productively rearranged group. In response to the Reviewer's concern we have softened our conclusion to read (lines 306-307): “*The reduced rearrangement of V_H11.2 in the NE1^{-/-} pro-B cells is therefore a **possible** explanation for the reduction of splenic B1a cells and those binding PtC. Thus, perturbation of the preselected Igh repertoire in pro-B cells has **potential** ramifications for the peripheral repertoire.*”

Bibliography

1. Dai, H.Q. *et al.* (2021) Loop extrusion mediates physiological Igh locus contraction for RAG scanning. *Nature* 590, 338-343. 10.1038/s41586-020-03121-7
2. Lee, Y.N. *et al.* (2014) A systematic analysis of recombination activity and genotype-phenotype correlation in human recombination-activating gene 1 deficiency. *J Allergy Clin Immunol* 133, 1099-1108. 10.1016/j.jaci.2013.10.007

3. Oudinet, C. *et al.* (2020) Recombination may occur in the absence of transcription in the immunoglobulin heavy chain recombination centre. *Nucleic Acids Res* 48, 3553-3566. 10.1093/nar/gkaa108
4. Andersson, R. and Sandelin, A. (2020) Determinants of enhancer and promoter activities of regulatory elements. *Nat Rev Genet* 21, 71-87. 10.1038/s41576-019-0173-8
5. Heintzman, N.D. *et al.* (2007) Distinct and predictive chromatin signatures of transcriptional promoters and enhancers in the human genome. *Nat Genet* 39, 311-318. ng1966 [pii] 10.1038/ng1966
6. Creighton, M.P. *et al.* (2011) Histone H3K27ac separates active from poised enhancers and predicts developmental state. *Proc Natl Acad Sci U S A* 107, 21931-21936. 1016071107 [pii] 10.1073/pnas.1016071107
7. Raisner, R. *et al.* (2018) Enhancer Activity Requires CBP/P300 Bromodomain-Dependent Histone H3K27 Acetylation. *Cell Rep* 24, 1722-1729. 10.1016/j.celrep.2018.07.041